# Structural and biochemical comparison of the FLVCR and CTL membrane protein families in eukaryotes

Lynette Nel*, Jan H Driller*, Ronja Driller, Kelly M Frain, Bjørn P Pedersen

The organic cation choline is essential for eukaryotic metabolism. Recently, the feline leukemia virus subgroup C receptor–related (FLVCR, SLC49) family was demonstrated as central for basal choline transport, questioning the role of the choline transporter-like (CTL, SLC44) family in this capacity. Here, we use *Xenopus laevis* oocytes to confirm that FLVCR1 (SLC49A1) and FLVCR2 (SLC49A2) proteins are choline transporters. CTL1 (SLC44A1) does not transport choline under the same conditions, supported by other CTL proteins, *Arabidopsis thaliana* CherI and *Saccharomyces cerevisiae* PNS1, which also display no choline transport activity. We present the atomic structures of FLVCR2, CTL1, and PNS1. The 3.4 Å cryo-EM structure of FLVCR2 has choline in the binding pocket. The 3.3 Å cryo-EM structure of CTL1 and the 2.7 Å crystal structure of PNS1 reveal an unusual protein fold, weakly related to the mitochondrial carrier family (SLC25). The unusual fold appears incompatible with transmembrane transport and implies a different and, so far, unknown function for CTL proteins. Our results support FLVCR proteins as choline transporters and suggest a nontransport role for CTL proteins.

## Introduction

Choline is an essential nutrient for humans and acts as a precursor for the neurotransmitter acetylcholine and membrane lipids such as phosphatidylcholine (PC) and sphingomyelin through the Kennedy pathway (1, 2, 3, 4, 5) (Fig 1A). Dysregulation of choline homeostasis has been implicated in numerous and diverse illnesses, for example, inflammatory bowel disease, Alzheimer's disease, coronary artery disease, and neonatal respiratory distress syndrome (6, 7, 8, 9). More broadly, choline holds a central position in the metabolism of all eukaryotes, including methionine and purine biosynthesis (Fig 1A) (1, 3, 6, 10).

Because choline is an organic cation, with a concentration of ~10 $\mu$M in the human bloodstream, a basal level of choline uptake needs to be facilitated by transport proteins for cells to function optimally (11, 12, 13, 14). So far, at least five protein families have been implicated in cellular choline uptake in humans (Fig 1B): the high-affinity choline transporter family (CHT, SLC5) (15), the organic cation transporter family (OCT, SLC22) (16), the mitochondrial carrier family (MCF, SLC25) (10), the choline transporter-like family (CTL, SLC44) (17), and the feline leukemia virus subgroup C receptor–related family (FLVCR, SLC49) (18).

As the name suggests, the SLC5 high-affinity choline transporter family contains high-affinity transporters, which are only expressed in cholinergic neurons and are essential for acetylcholine synthesis but not basal choline uptake (19). Human CHT1 (SLC5A7) has a $K_m$ of 0.5–5 $\mu$M (19, 20). The SLC22 organic cation transporters are suggested to be lower affinity ($K_m$ ~1.5 mM) choline transporters, but are also selectively expressed, and not thought to be responsible for basal-level choline uptake (21). In mitochondria, SLC25A48 has been identified as the major choline transporter, but as it is specific for mitochondria, this protein also cannot be responsible for basal choline uptake (10).

Since the early 2000s, it was suggested that the SLC44 choline transporter-like protein family was driving basal choline uptake in humans (4, 22). The protein family was initially identified using a *Saccharomyces cerevisiae* strain with a knockout of the gene "*hyper-resistant to nitrogen mustard*" (*hnm1*), the only known choline transporter in yeast. Using a cDNA library from *Torpedo marmorata* in a complementation study, a membrane protein named *T. marmorata* choline transporter-like, tCtl1p, was proposed to mediate high-affinity choline transport (22, 23). This transporter typified a new protein family, the CTL family, which includes human CTL1 (SLC44A1), *S. cerevisiae* PNS1 (pH nine-sensitive protein 1), and *Arabidopsis thaliana* CherI (Choline Transporter-like 1) (23, 24). However, soon after, another study cast some doubt on this proposition, showing that loss of PNS1 has no effect on choline transport in yeast and overexpression of either PNS1 or tCTL1 did not restore uptake of choline by transport-defective yeast mutants (25). Nevertheless, a range of studies later suggested the involvement of CTL proteins in choline and ethanolamine uptake (e.g., references (24, 26, 27, 28, 29, 30, 31)). The structure of human CTL1 as a dimer was published in 2022 and revealed a novel membrane protein fold, with a large, diffuse

Department of Molecular Biology and Genetics, Aarhus University, Aarhus, Denmark

Correspondence: bpp@mbg.au.dk
*Lynette Nel and Jan H Driller contributed equally to this work

 

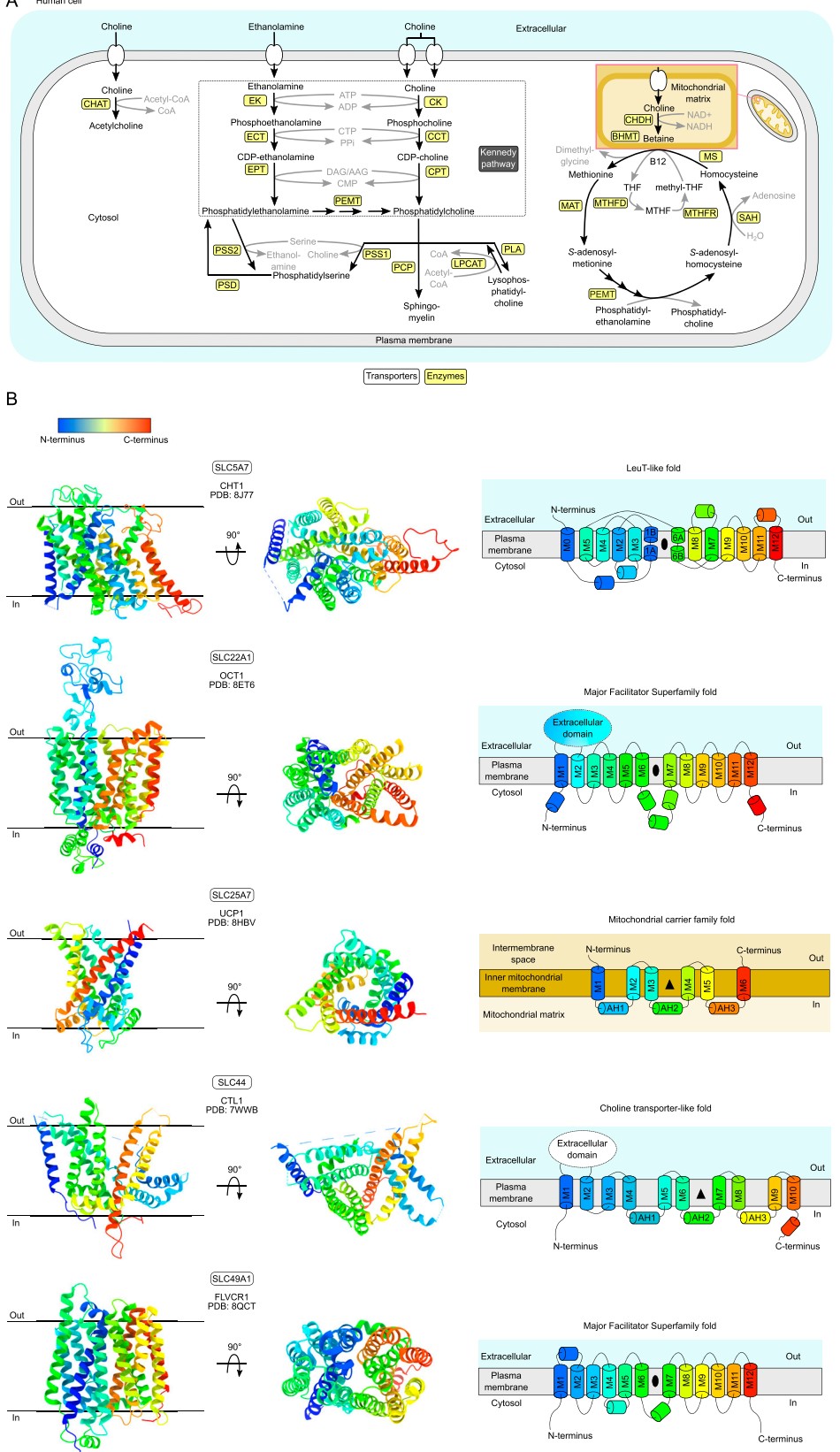

**Figure 1. Overview of the metabolism and transporters of choline and ethanolamine in humans.**

**(A)** Choline and ethanolamine need to be transported across biological membranes to be metabolized further by multiple enzymes. For instance, choline can be acetylated to produce the neurotransmitter acetylcholine. Both choline and ethanolamine can be phosphorylated in the Kennedy pathway to form phospholipids, which are essential components of all membranes. Within mitochondria, the oxidation of choline forms betaine, a precursor to amino acids like methionine and cysteine. Abbreviations: CHAT, choline acetyltransferase; EK, ethanolamine kinase; ECT, phosphoethanolamine cytidylyltransferase; EPT, ethanolaminephosphotransferase; CK, choline kinase; CCT, phosphocholine cytidylyltransferase; CPT, cholinephosphotransferase; PEMT, phosphatidylethanolamine *N*-methyltransferase; PSS1, phosphatidylserine synthase 1; PSS2, phosphatidylserine synthase 2; PCP, phosphatidylcholine:ceramide choline phosphotransferase; LPCAT, acyl-CoA: lysophosphatidylcholine acyltransferase; PSD, phosphatidylserine decarboxylase; PLA, platelet-activating factor; CHDH, choline dehydrogenase; BMHT, betaine:homocysteine methyltransferase; MTHF, 5,10-methylene-tetrahydrofolate; THF, tetrahydrofolate; MTHFD, methylenetetrahydrofolate dehydrogenase; MTHFR, methylenetetrahydrofolate reductase; MAT, methionine adenosyltransferase; SAH, *S*-adenosylhomocysteine hydrolase; and MS, methionine synthase. **(B)** Experimentally solved structures representing the human solute carrier (SLC) families implicated in choline transport. Models and topology diagrams are colored from N terminus (blue) to C terminus (red). The high-affinity choline transporter family (CHT, SLC5) has a LeuT-like fold and is exemplified by CHT1 (PDB: 8J77). Both the organic cation transporter family (OCT, SLC22) and the feline leukemia virus subgroup C receptor–related (FLVCR, SLC49) family have a major facilitator superfamily (MFS) fold. However, OCT1 (PDB: 8ET6) has an extracellular domain between TM1 and TM2 that FLVCR1 (PDB: 8QCT) does not have. Two unique but similar folds are typified by the mitochondrial carrier family (MCF, SLC25), represented by uncoupling protein 1 (UCP1, PDB: 8HBV), whereas the choline transporter-like (CTL, SLC44) family is characterized by CTL1 (PDB: 7WWB). Because no experimentally solved structure of SLC25A48 is available, we use the structure of uncoupling protein 1 (SLC25A7) to represent the fold.

density that was tentatively modeled as a smaller choline molecule in each monomer (17).

In parallel, the SLC49 protein family, previously identified as having a role in mediating feline leukemia virus subgroup C infection and believed to be heme transporters, was proposed to be responsible for basal-level choline uptake (32, 33). It was shown that human FLVCR2 (SLC49A2; MFSD7c) can restore choline uptake in *hnm1*-deficient *S. cerevisiae* (33). FLVCR2 was proposed to function predominantly as a bidirectional choline uniporter and is highly expressed at the blood–brain barrier to export excessive choline from lysophosphatidylcholine and PC metabolism (34). At the same time, human FLVCR1 (SLC49A1) was shown to transport choline but suggested to preferentially transport ethanolamine, another precursor of phospholipid biosynthesis (Fig 1A) (18, 35). Since 2024, there has been a large effort to biochemically and structurally characterize human FLVCR proteins, leading to 13 different deposited structures of FLVCR1 and FLVCR2, several with either choline or ethanolamine bound in a central binding pocket (18, 33, 35).

Here, we use *Xenopus laevis* oocyte uptake assays, X-ray crystallography, and cryo-EM to assess CTL and FLVCR proteins. In comparative transport assays, we do not detect choline transport by CTL1, PNS1, or CherI, whereas we observed choline transport by FLVCR1 and FLVCR2. We present the structures of human FLVCR2 (cryo-EM, 3.4 Å) and human CTL1 (cryo-EM, 3.3 Å) and the first structure of *S. cerevisiae* PNS1 (X-ray, 2.7 Å). We can confidently say that FLVCR2 has choline bound, whereas the maps of CTL1 and PNS1 reveal a larger unidentified ligand, not compatible with choline. It is still unclear what the function of CTL1 and PNS1 is, and their fold is curiously unique. The fold of CTL1 and PNS1 weakly resembles the fold of SLC25 proteins, but it is not yet evident whether the architecture of CTL proteins would allow the same or similar mechanism of transport as SLC25 members.

## Results

### Biochemical characterization of members of the FLVCR and CTL protein families

We tested choline and ethanolamine transport using radiolabeled substrate uptake assays in *X. laevis* oocytes (Fig S1A). We assayed three CTL proteins (*Homo sapiens* CTL1, *S. cerevisiae* PNS1, and *A. thaliana* CherI) and two FLVCR proteins (*H. sapiens* FLVCR1 and *H. sapiens* FLVCR2) in parallel, using an identical experimental setup. Confocal microscopy was used to verify the expression of each GFP-tagged protein on the plasma membrane of oocytes (Fig S1B).

Transport of both choline and ethanolamine is observed for FLVCR1 and FLVCR2 (Fig 2A and B). In contrast, despite extensive efforts, no uptake of choline or ethanolamine could be established for CTL1, PNS1, and CherI (Fig 2A and B).

We next determined the apparent affinity ($K_m$) of FLVCR1 and FLVCR2 for choline as ~26 $\mu M$ and ~164 $\mu M$, respectively (Fig 2C and D). Although linear uptake of choline was observed for FLVCR1 and FLVCR2 (Fig S1C), saturation kinetics could not be

obtained, despite adding up to 2 mM ethanolamine, which is 1,000-fold above the physiological concentration in blood of ~2 $\mu M$ (Fig S1D) (14, 36). To expand on the characterization of the substrate scope of FLVCR1 and FLVCR2 suggested in literature (18, 34), we next tested radiolabeled carnitine and histamine because these compounds are chemically related to choline and ethanolamine, respectively. However, neither FLVCR1 nor FLVCR2 transports carnitine or histamine in our experimental setup (Fig 2E). We also tested radiolabeled betaine, a derivative of choline, but we could not obtain useful data, as the membrane integrity of oocytes was compromised in the presence of this compound.

Next, we tested pH dependence of choline transport. In a range from pH 5.5 to pH 8.5, CTL1, PNS1, and CherI did not transport 0.5 $\mu M$ or 2 $\mu M$ choline (Fig S1E and F). When using 50 $\mu M$ choline, FLVCR1 activity seems independent of pH (in pH range 6.0–8.5), whereas FLVCR2 displays some pH dependency in the same range, with reduced transport activity at pH 6.0 (Fig 2F). We also tested whether transport is dependent on the membrane potential using the proton decoupler carbonyl cyanide 3-chlorophenylhydrazone (CCCP). The transport of choline by FLVCR1 is not significantly reduced by the addition of CCCP (Fig 2G). For FLVCR2, however, the transport of choline is indeed significantly decreased when CCCP is added (Fig 2G). Lastly, we investigated the role of sodium in choline transport, as part of a possible sodium-driven symport mechanism. We found that the transport of choline and ethanolamine by FLVCR1 and FLVCR2 is not affected by the presence or absence of sodium (Fig 2H).

### Cryo-EM structure of *H. sapiens* FLVCR2 with choline bound

FLVCR proteins belong to the major facilitator superfamily (MFS), topologically consisting of 12 transmembrane helices that are arranged in an N domain (M1–M6) and a C domain (M7–M12) around a central twofold symmetry point (Fig 3A) (18, 35). We determined the structure of FLVCR2 in *n*-dodecyl-b-maltoside (DDM) detergent to 3.4 Å using cryo-EM (Figs 3B and S2A–F, Table S1). Because of the small size and twofold pseudosymmetry of this MFS transporter, a repeat of three Protein A sequences was fused to the N terminus of FLVCR2 to serve as a fiducial marker and facilitate particle alignment (37). When superposing our FLVCR2 model with a model prediction of Protein A fused to FLVCR2, we can see that the Protein A repeat is approximately in the same position as seen in the map (Fig 3C). The map shows elongated densities surrounding FLVCR2, possibly cholesterol-hemisuccinate (CHS) molecules or retained lipids (Fig 3D). FLVCR2 is captured in an inward-facing conformation, with clear density for choline in the central binding site at the pseudosymmetry point between the N and C domains (Fig 3E). The hydroxyl group of choline is coordinated by Gln447(M11), whereas the methyl groups are oriented toward Gln191(M4) and Ile226(M5) (Fig 3E). Trp102(M1) and Phe324(M7) obstruct passage to the extracellular side. The cavity leading to the binding pocket of FLVCR2 has mostly hydrophobic and negatively charged residues (Fig 3F and G). On the outer surface, FLVCR2 has hydrophobic and positively charged patches that correspond to the locations where lipids and sterols might bind (Fig 3D, H, and I).

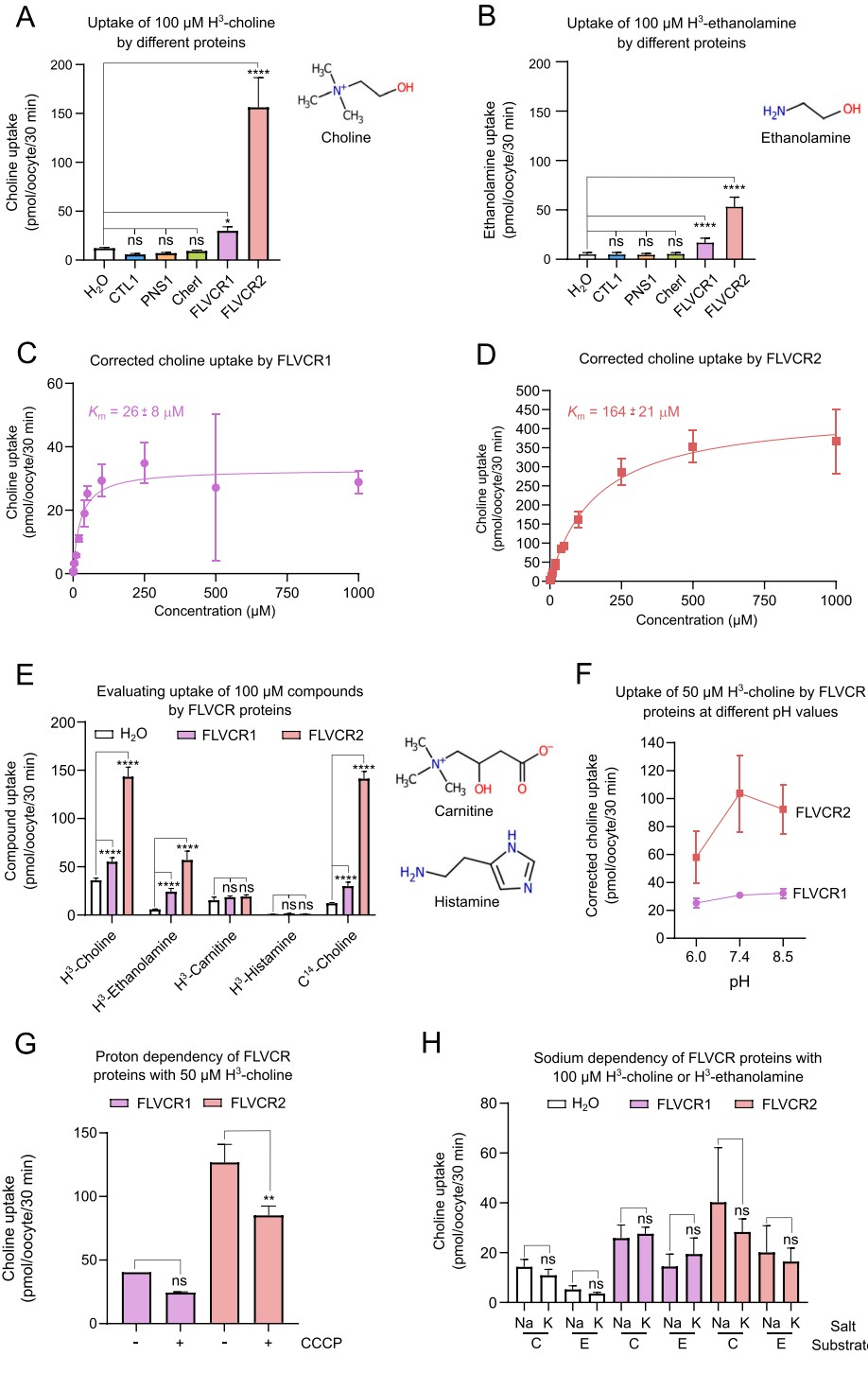

**Figure 2. Biochemical characterization of proposed choline transporters in *Xenopus laevis* oocytes.**

Statistical significance is denoted by asterisks that correspond to certain probabilities (*P*-values). Not significant (ns; *P* > 0.05), significant (**P* < 0.05), highly significant (***P* < 0.01), very high significance (****P* < 0.001), and extremely high significance (*****P* < 0.0001). **(A)** Uptake of 100 $\mu$M H$^3$-choline by CTL1, PNS1, CherI, FLVCR1, and FLVCR2 (n = 4–6). Compared with the negative control of water, choline uptake is not observed for CTL1 (*P* = 0.7376), PNS1 (*P* = 0.8621), or CherI (*P* = 0.9875). Choline uptake is observed for FLVCR1 (*P* = 0.0208) and FLVCR2 (*P* < 0.0001). Statistics were calculated from a one-way ANOVA with Dunnett's multiple comparisons test. **(B)** Uptake of 100 $\mu$M H$^3$-ethanolamine by CTL1, PNS1, CherI, FLVCR1, and FLVCR2 (n = 6). Compared with the negative control of water, ethanolamine uptake is not observed for CTL1 (*P* > 0.9999), PNS1 (*P* = 0.9977), or CherI (*P* > 0.9999). Ethanolamine uptake is observed for FLVCR1 (*P* < 0.0001) and FLVCR2 (*P* < 0.0001). Statistics were calculated from a one-way ANOVA with Dunnett's multiple comparisons test. **(C)** Titration curve of FLVCR1 with H$^3$-choline (n = 4–6). Measurements were corrected by subtracting the average of counts obtained from water for each concentration. A Michaelis–Menten fit was applied to obtain the apparent affinity ($K_m$) for FLVCR1 as 26 ± 8 $\mu$M. **(D)** Titration curve of FLVCR2 with H$^3$-choline (n = 3–6). Measurements were corrected by subtracting the average of counts obtained from water for each concentration. A Michaelis–Menten fit was applied to obtain the apparent affinity ($K_m$) for FLVCR2 as 164 ± 21 $\mu$M. **(E)** Evaluating the substrate scope of FLVCR1 and FLVCR2 with choline, ethanolamine, carnitine, and histamine (n = 2–6). When comparing water with H$^3$-choline, FLVCR1 (*P* < 0.0001) and FLVCR2 (*P* < 0.0001) exhibited transport. For water compared with H$^3$-ethanolamine, FLVCR1 (*P* < 0.0001) and FLVCR2 (*P* < 0.0001) showed transport. When comparing water with H$^3$-carnitine, FLVCR1 (*P* = 0.9090) and FLVCR2 (*P* = 0.7424) did not exhibit transport. For water compared with H$^3$-histamine, FLVCR1 (*P* > 0.9999) and FLVCR2 (*P* > 0.9999) did not show transport. When comparing water to C$^{14}$-choline, FLVCR1 (*P* < 0.0001) and FLVCR2 (*P* < 0.0001) exhibited transport. Statistics were calculated from a one-way ANOVA with Sidak's multiple comparisons test. **(F)** Transport of H$^3$-choline by FLVCR1 and FLVCR2 under different pH conditions (n = 3–4). Measurements were corrected by subtracting the average of counts obtained from water for each pH condition. **(G)** Influence of protons and the membrane potential on the transport of choline by FLVCR1 and FLVCR2 (n = 2–4). When comparing FLVCR1 without and with CCCP (*P* = 0.2283), no significant dependency on protons was observed. For FLVCR2 with and without CCCP (*P* = 0.0011), a highly significant dependency on protons was observed. Statistics were calculated from a one-way ANOVA with Sidak's multiple comparisons test. **(H)** Transport of H$^3$-choline (C) and H$^3$-ethanolamine (E) by FLVCR1 and FLVCR2 in the presence (Na) of 96 mM sodium or absence (K) of sodium (using 96 mM potassium; n = 3–5). Neither FLVCR1 nor FLVCR2 seems to depend on sodium to transport choline or ethanolamine. For negative controls, water and choline with sodium and with potassium, *P* = 0.9801, as well as water and ethanolamine with sodium or potassium, *P* = 0.9996 were obtained. For FLVCR1 with choline and sodium or potassium, *P* = 0.9999. For FLVCR1 and ethanolamine and sodium or potassium, *P* = 0.9152. For FLVCR2 with choline and sodium or potassium, *P* = 0.1761. For FLVCR1 and ethanolamine and sodium or potassium, *P* = 0.9795. Statistics were calculated from a one-way ANOVA with Sidak's multiple comparisons test.

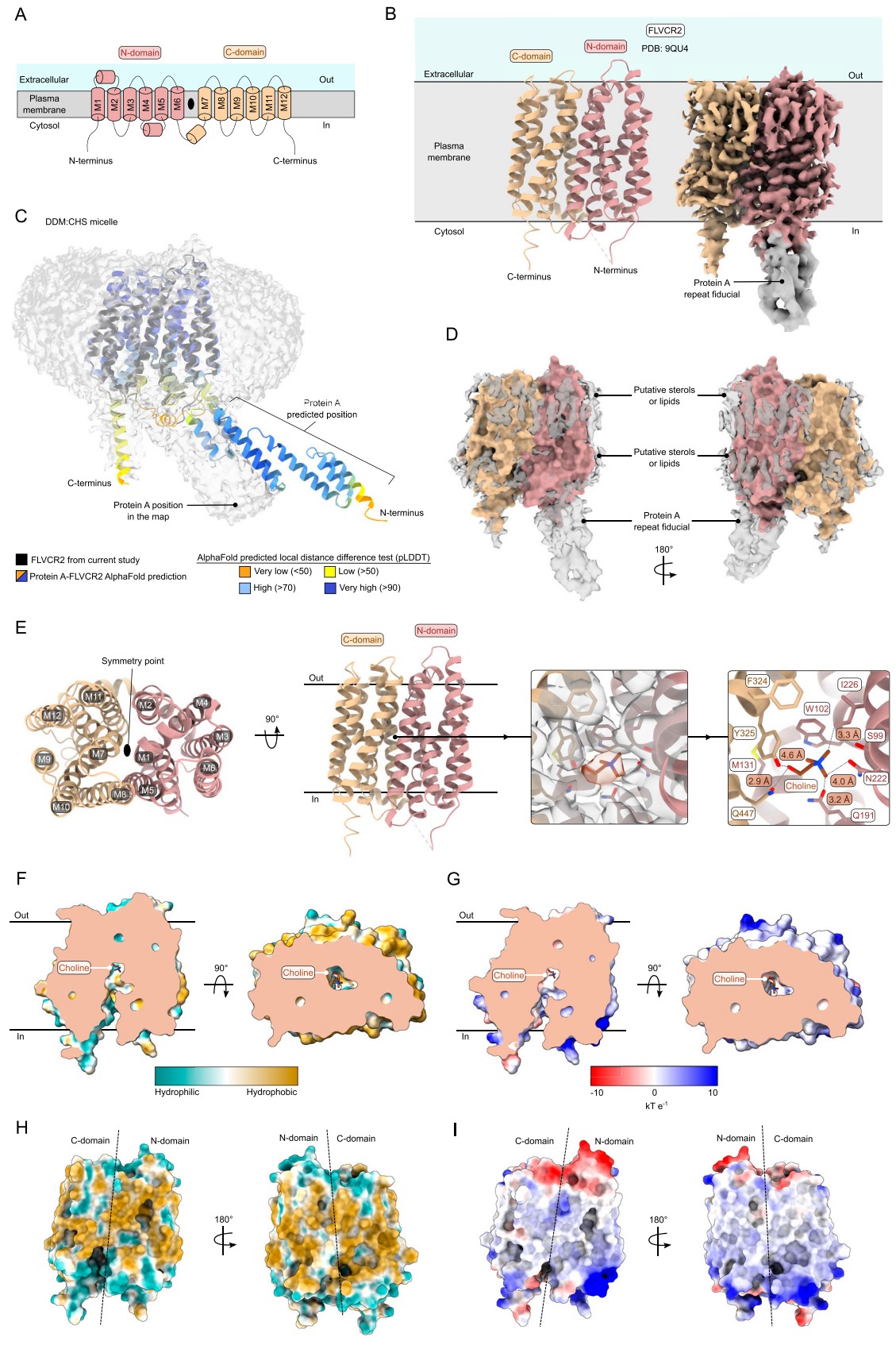

## Cryo-EM structure of *H. sapiens* CTL1

We expressed, purified, and determined the 3.3 Å cryo-EM structure of human CTL1 reconstituted in lauryl maltose neopentyl glycol (LMNG) detergent and CHS (Fig S3A–F and Table S1). In accordance with previous results (17), we observe a unique and highly unusual fold for CTL1 among SLC proteins, with 10 transmembrane helices (M1–M10), a large extracellular domain bridging M1 with M2, and three shorter amphipathic helices (AH1-3) parallel to the membrane that connect M4–M5, M6–M7, and M8–M9, respectively (Fig 4A). As a control for detergent solubilization–mediated artifacts, we prepared a cryo-EM sample in peptidisc and determined the map to 3.3 Å (EMDB: EMD-50251). The core of the volumes is the same with the only noticeable difference in the positioning of M3/M4, which is compressed in the map solved with peptidisc (map–map cross-correlation coefficient = ~0.9; Fig S3G); thus, we proceeded with the analysis using the sample in LMNG. CTL1 is present as a monomer and is overall reminiscent of a bowl-like structure embedded in the membrane, with the opening facing toward the extracellular side. The map regions covering the extracellular domain (residues 63–213) and most of M3/M4 (residues 248–305) were very poorly defined, and we left these sections of the protein unmodeled (Fig 4B). When superposing a predicted model of full-length CTL1 on our experimentally determined CTL1 model, the prediction helps to approximate where unmodeled regions of the extracellular domain and M3/M4 protrude from the micelle, matching the map when displayed at a lower contour level (Fig 4C). Multiple elongated densities, possibly sterols or retained lipids, can be seen surrounding M1, M5, and M6, at the interface with M7, and in a hydrophobic pocket at the back of the protein, formed by M1, M5, and M6 (Fig 4D). At the center of the threefold pseudosymmetry point, corresponding to the location of the putative binding pocket suggested by Xie and coworkers, an electrostatic potential for a ligand appears at low contour level but, as they noted and we confirm here, it is too large to correspond to stably bound choline (Fig 4E) (17). The residues defining the pocket are Gln404(M6), Lys491(M8), Asn494(M8), Tyr495(M8), Gln498(M8), Asn533(M9), Asp537(M9), Glu591(M10), and Met592(M10). The outer surface of CTL1 is mostly hydrophobic, with a groove at the back where we observe the elongated densities in the map, whereas the putative binding pocket largely consists of hydrophilic residues (Fig 4F). The outer surface of CTL1 is mostly neutral to positively charged, whereas distinct positive (Lys544, Lys454, Arg457, His584, and Lys491) and negative (Asp537, Glu591, and Asp595) patches are observed close to the entrance of the proposed binding pocket (Fig 4G).

## Crystal structure of *S. cerevisiae* PNS1

To verify the unusual SLC44 fold observed in CTL1, and hopefully improve our understanding of its function, we determined the structure of *S. cerevisiae* PNS1, the yeast homolog of CTL1, by lipidic cubic phase (LCP) crystallography to 2.7 Å and experimental phasing, ensuring a completely unbiased structural comparison (Fig S4A–D and Table S2). The protein crystallized in space group $P22_12_1$ with one molecule in the asymmetric unit. Experimental phases were obtained by single-wavelength anomalous diffraction (SAD) phasing using $KAu(CN)_4$. Like CTL1, PNS1 also contains 10 transmembrane helices (M1–10) forming a transmembrane bowl-like structure with M1 and M2 located at the periphery and four pairs of antiparallel helices (M3/4, M5/6, M7/8, and M9/10) (Fig 5A and B). Helices M4/M5, M6/M7, and M8/M9 are intersected by the amphipathic helices AH1-3 laying parallel to the membrane, creating a threefold pseudosymmetrical arrangement. The model is missing the unstructured N terminus of 76 amino acids, as well as several loops connecting the transmembrane helices (Fig 5B). Like CTL1, there seems to be elongated densities that could match lipids or sterols at the back of PNS1, opposite the entrance of the putative binding pocket (Fig 5C). Densities for water molecules and an unknown ligand, too large to be choline, are also observed within the proposed binding pocket in a similar position as for CTL1. (Fig 5D). As such, the putative ligand is left unmodeled also in PNS1, but the surrounding residues were identified as Arg296(M6), Trp390, Glu393, Tyr394, His397(M8), Asp432, Asn436(M9), Glu492, and Arg495(M10). Salt bridges could form between Arg296 and Glu393 (4 Å apart), as well as Glu492 and Arg495 (2.5 Å away), whereas a water molecule can mediate hydrogen bond formation between Arg296 and Glu492 (6.3 Å apart). Although the outer surface of PNS1 is mostly hydrophobic, the putative binding pocket is largely hydrophilic (Fig 5E). PNS1 is neutral to positively charged on the outer surface, but some distinct positively (Arg296, Arg354, Lys389, and Arg495) and negatively charged (Glu293, Glu393, and Glu392) patches are located within the proposed binding pocket (Fig 5F).

## CTL1 and PNS1 have an unusual membrane protein fold

The bowl-like opening of CTL1 is ~20 Å wide and ~30 Å deep, whereas PNS1 has a similar-sized cavity open to the extracellular side of ~26 Å wide and ~35 Å deep (Fig 6A). When aligning $CTL1_{AH1–AH3}$ to $PNS1_{AH1–AH3}$, the $RMSD_{Ca}$ is 1.1 Å, with AH2 and AH3 in the same position but AH1 of CTL1 is displaced downward compared with that of PNS1 (Fig 6B). This shift in AH1 allows M3/M4 to be more open in CTL1 and is the biggest conformational difference

---

**Figure 3. Cryo-EM structure of FLVCR2.**
**(A)** Diagram of the major facilitator topology exhibited by FLVCR2, divided into an N domain (rosy brown) and C domain (sand). The twofold pseudosymmetry point is indicated by the black oval. **(B)** Model and map of FLVCR2 (PDB: 9QU4) in a schematic of its localization in the plasma membrane. The map shows where the Protein A repeats (gray) were linked to the N terminus, but this fiducial was not built in the model. **(C)** Map of the DDM:CHS micelle, shown as a gray surface, with our model of FLVCR2 inside in black. The AlphaFold2 prediction of Protein A fused to FLVCR2, colored according to the predicted local distance difference test (pLDDT) scale, is superposed onto our model. **(D)** Unmodeled, elongated map regions, possibly of sterols and retained lipids, form belts around FLVCR2, particularly on the face where the membrane would be in contact with the extracellular environment. **(E)** Model of FLVCR2 at 3.4 Å adopts the inward-facing conformation. Choline (brown) is bound within a pocket located at a central twofold pseudosymmetry point. Residues involved in coordinating choline are shown as sticks and annotated in the map (gray). **(F)** Hydrophobicity of the binding pocket with choline inside. The slices were made through the front and the top of FLVCR2. **(G)** Electrostatics of the binding pocket with choline inside. The slices were made through the front and the top of FLVCR2. **(H)** Hydrophobicity coloring of FLVCR2 from the front and the back. **(I)** Electrostatic coloring of FLVCR2 from the front and the back.

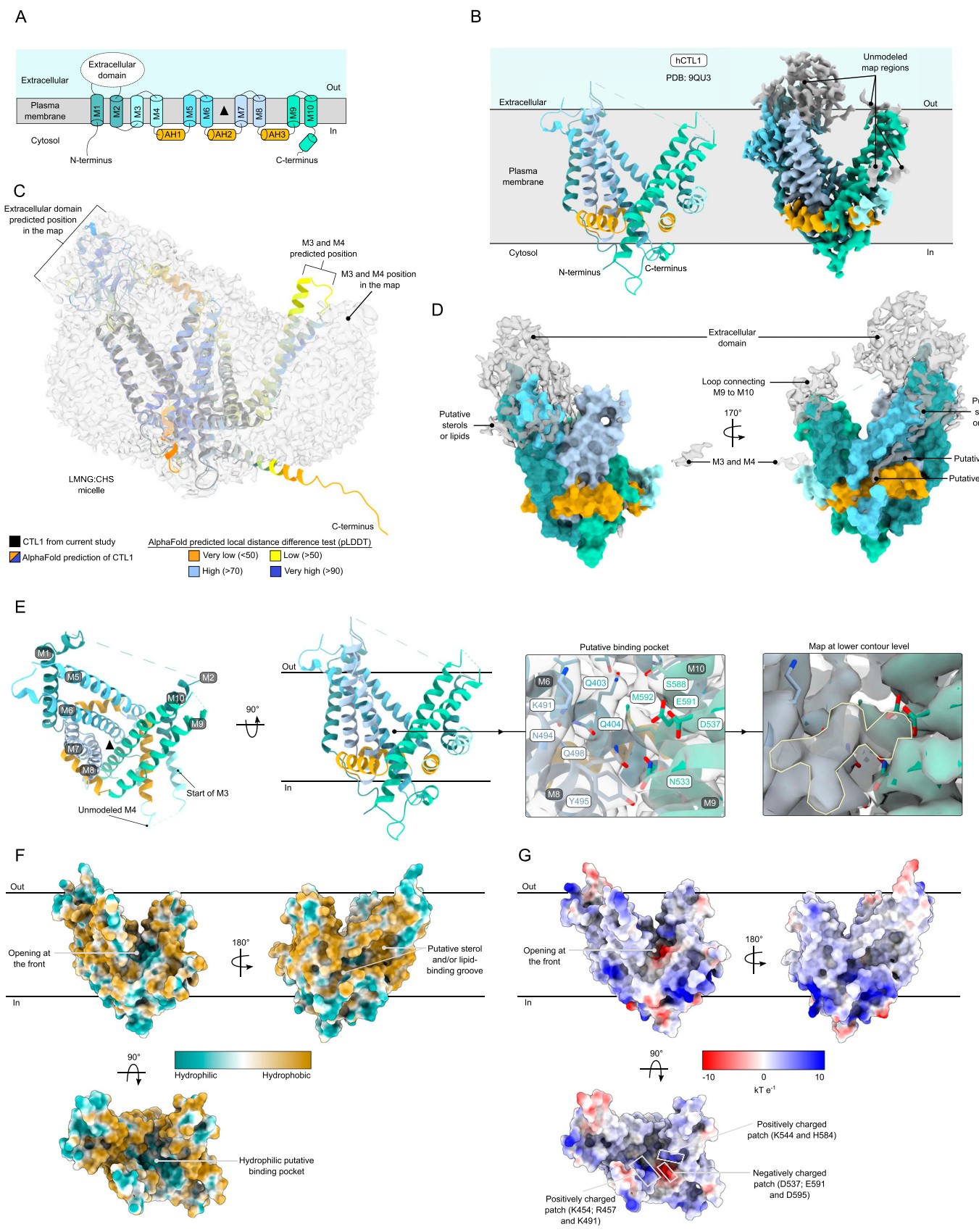

in relation to PNS1. The core helices (M5/6, M7/M8, and M9/M10) are in the same position at the putative binding pocket, but their placement diverges toward the top of the structures (Fig 6B). PNS1 and CTL1 share 15% sequence identity, with only Tyr495$_{CTL1}$ and Tyr394$_{PNS1}$ conserved in the putative binding pocket (Fig 6C), and, intriguingly, most of the conserved residues form the hydrophobic groove at the back, where the putative sterol/lipid densities are observed (Fig 6D). When looking at the evolutionary conservation of residues, both CTL1 and PNS1 have conserved residues in the proposed binding pocket and the amphipathic helices parallel to the membrane (Fig 6E). To investigate the binding site of the CTL family further, we added hemicholinium-3 (HC-3), a competitive inhibitor of CHT1 (15), to both the CTL1 cryo-EM sample and PNS1 crystallization condition, in the hope that it would also function as an inhibitor for these proteins, but the maps did not reveal the presence of HC-3 or altered conformations for either CTL1 or PNS1. Thus, the PNS1 structure confirms the unique membrane protein fold of human CTL1 but unfortunately does not shed new light on the mechanistic properties of the SLC44 family, nor its physiological function.

# Discussion

The current study investigates the activity and substrate scope of FLVCR proteins and compares them with the CTL proteins that, with albeit contradictory results, were previously proposed to be choline transporters in the literature.

We show that FLVCR1 and FLVCR2 proteins from humans can transport choline in *Xenopus* oocytes, supporting past findings in a new experimental system. We determine an apparent affinity ($K_m$) of ~26 $\mu$M for FLVCR1, which lies in-between the values observed by Ri and coworkers (~47 $\mu$M) and Son and coworkers (~8 $\mu$M), who used HEK cell assays (18, 35). The $K_m$ of FLVCR2 for choline is around 164 $\mu$M, which is ~2.5 times weaker compared with that obtained by Ri and coworkers ($K_m$ ~64 $\mu$M) in HEK cells (18). This stronger apparent affinity in HEK cells could be due to the FLVCR proteins being co-expressed with choline kinase A, leading to a decrease in the local concentration of choline by shunting the equilibrium toward phosphocholine (18). Another study by Nguyen and coworkers also co-expressed FLVCR2 with choline kinase A in HEK cells and obtained a $K_m$ of around 100 $\mu$M (33), which is a value comparable to the result we obtained. In this study, we see that FLVCR1 shows about six times stronger apparent affinity for choline

compared with FLVCR2. Notably, this affinity is still 2.5× higher than the known choline concentration in blood plasma (~10 $\mu$M) (12, 14). Overall, differences in $K_m$ values between various experimental setups are expected, which might explain the differences observed in this work compared with others (38).

In oocytes, both FLVCR1 and FLVCR2 display ethanolamine uptake, but without saturation kinetics. For FLVCR1, this is in contrast with previous studies where it was reported to transport ethanolamine with a $K_m$ of ~3–8 $\mu$M, having a two to five times higher affinity for ethanolamine compared with choline (18, 35). Notably, this reported affinity would still be higher than the known ethanolamine concentration in blood plasma (~2 $\mu$M) (14, 36). Compared with these previous studies, our results indicate that both FLVCR1 and FLVCR2 prefer choline as a substrate to ethanolamine. One possible contributing factor for this discrepancy could be that several of the fitted kinetic curves obtained in HEK cells do not seem to reach saturation for either choline or ethanolamine under the published assay conditions. Again, these discrepancies could also be explained by changes in the experimental setup because apparent affinities are influenced by the system that is used (38).

One previous study investigated carnitine as a possible substrate for FLVCR2 in HEK cells and found that if FLVCR2 was co-expressed with human choline kinase A, carnitine uptake slightly increased (33). Cater and coworkers showed that murine FLVCR2 has comparable affinity for acetylcholine, betaine, and choline in HEK cell uptake assays (34). In contrast, Ri and colleagues found that betaine arrested both human FLVCR1 and FLVCR2 in a way that did not allow transport of choline and speculated that it could be because of a hydroxyl-to-carbonyl substitution between choline and betaine (18). Our study shows that human FLVCR1 and FLVCR2 preferentially transport choline and possibly ethanolamine and do not transport the chemically related compounds carnitine and histamine in oocytes.

We find that choline transport by FLVCR1 is independent of pH, whereas FLVCR2 transports optimally at pH 7.4 and exhibits reduced activity when the proton gradient is abolished by CCCP. Cater and colleagues investigated the driving force underlying choline transport with murine FLVCR2 in proteoliposomes, which seems to prefer a range of neutral to basic pH (pH 7.5–9.5) to transport choline. The authors proposed that protons could have a regulatory role in the uniport mode of choline transport (34). This is compatible with our observations for human FLVCR2; when the proton gradient is disrupted by CCCP, choline transport is significantly reduced. This suggests that FLVCR2 operates by either

**Figure 4.  Cryo-EM structure of CTL1.**
**(A)** Diagram of the novel topology of CTL1. The threefold pseudosymmetry point is indicated by the black triangle. **(B)** Model and map of CTL1 (PDB: 9QU3) in a schematic of its localization in the plasma membrane. The model of CTL1 shows a unique fold with 10 transmembrane helices (M1–10 in shades of blue and green) arranged in a rotary fashion around three amphipathic helices (AH1–3 in yellow) laying parallel to the membrane. The extracellular domain and M3 and M4 were not built and are shown in gray in the map. **(C)** Map of the LMNG:CHS micelle, shown as a gray surface, with our model of CTL1 inside in black. The AlphaFold2 prediction of CTL1, colored according to the predicted local distance difference test (pLDDT) scale, is superposed onto our model. **(D)** Unmodeled map regions of putative sterols and lipids form a belt around the transmembrane helices that are near the extracellular domain. Lipid- and sterol-like densities are also observed in a defined groove at the back of CTL1. **(E)** Model of CTL1 adopts a bowl-like conformation that is open toward the extracellular environment, with a possible binding pocket at the central pseudosymmetry point. Residues of the putative binding pocket are annotated and shown in the map (gray), along with an unidentified ligand (yellow outline) at a lower contour level of the map. **(F)** Hydrophobicity coloring of CTL1 shown from the front, back, and top. The outer surface of CTL1 is mostly hydrophobic, with a groove at the back where lipids or sterols likely bind, whereas the putative binding pocket mostly contains hydrophilic residues. **(G)** Electrostatic coloring of CTL1 shown from the front, back, and top. The outer surface of CTL1 is mostly neutral to positively charged, whereas distinct positive (Lys544, Lys454, Arg457, His584, and Lys491) and negative (Asp537, Glu591, and Asp595) patches are observed near the opening of the proposed binding pocket.

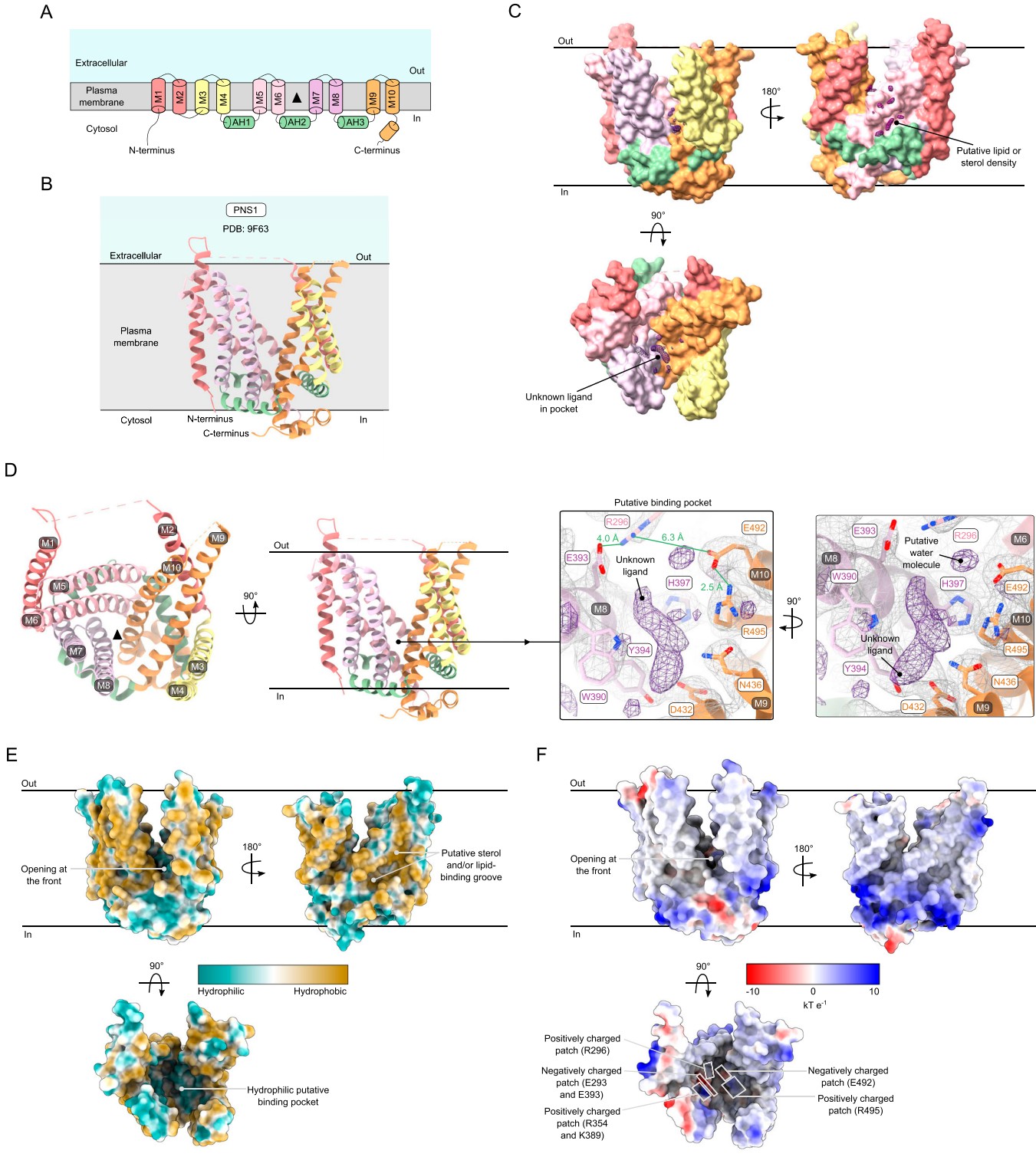

**Figure 5. X-ray crystallography structure of PNS1.**
**(A)** Diagram of the topology of PNS1. The threefold pseudosymmetry point is indicated by the black triangle. **(B)** Model of PNS1 in a schematic of its localization in the plasma membrane. PNS1 at 2.7 Å (PDB: 9F63) shows a novel fold with 10 transmembrane helices (M1-10) arranged in a rotary fashion (coral, yellow, pink, purple, and orange) and three amphipathic helices (AH1-3 in green) that lay parallel to the membrane. **(C)** Densities of lipid- or sterol-like molecules (purple mesh) are present in a defined groove at the back of PNS1. Within the putative binding pocket, densities for water molecules and an unknown ligand are also observed (purple mesh). **(D)** Model of PNS1 adopts a bowl-like conformation that is open toward the extracellular environment, with a possible binding pocket at the central pseudosymmetry point. Residues of the putative binding pocket are annotated and shown in the map (gray mesh), along with nonprotein densities attributed to an unidentified ligand and possible water molecules (purple mesh). Distances (green) are shown between residues that could form salt bridges or interactions with water molecules. **(E)** Hydrophobicity coloring of PNS1 shown from the front, back, and top. The outer surface of PNS1 is mostly hydrophobic, with a groove at the back where lipids or

proton-driven symport or uniport, as choline is positively charged ([39], [40]).

We find FLVCR1 and FLVCR2 are not dependent on sodium to transport choline and ethanolamine. These findings align with assays in HEK cells and proteoliposome assays in other studies ([18], [35]). Overall, our results support that human FLVCR proteins operate independently of sodium and can facilitate transport of positively charged choline across a membrane, along its electrochemical gradient.

In the past few years, there has been a large effort to characterize FLVCR proteins by solving structures of inward and outward open states of both FLVCR1 and FLVCR2 ([18], [34], [35]). We determined the inward-facing conformation of FLVCR2, assisted by an N-terminal 3-repeat Protein A fiducial, in DDM detergent to 3.4 Å (Figs 3 and S2). By superposing an AlphaFold-predicted model of the FLVCR2–Protein A chimera onto our model, it can be visualized that the protruding fiducial worked as intended (Fig 3C). The residues lining the binding pocket of human FLVCR2 are hydrophobic at the top (Trp102, Ile226, and Phe324) and mostly polar at the bottom and the sides (Ser99, Gln191, Asn222, Tyr325, Gln447). Among human and murine FLVCR2 inward open structures, choline is bound in the same position and the residues of the binding pocket have the same rotamers, except for one nonconserved residue (Ser99 in humans and Ala122 in mice) (Fig S5A). When superposing our structure to other structures of FLVCR2 in the inward conformation, the $RMSD_{Ca}$ is less than 1 Å ([18], [34]). These structures were obtained either without the use of a fiducial or using an antibody or, in our case, Protein A repeats. Even though our structure does not provide new information on FLVCR2, it confirms previous findings regarding choline binding and it supports that the use of Protein A did not perturb FLVCR2 in adopting the inward conformation, making it suitable for solving structures of small membrane proteins with pseudosymmetry and no discernible features protruding from the micelle.

The choline transporter-like family, CTL, was first identified based on complementation experiments of CTL from *T. marmorata* in budding yeast, with a reported choline uptake affinity of around 1 μM ([22]). Since then, the CTL family has been expanded to five members (CTL1–CTL5) in humans ([4]). Several studies addressing choline uptake have been conducted using CTL proteins—under the assumption that they are involved in the transport of choline or choline derivatives—using in vivo and recombinant experimental formats. For example, follow-up *X. laevis* oocyte uptake experiments suggested that *Tm*CTL was a sodium-independent transporter of choline ([29]). From radiolabeled choline uptake experiments in lung cancer cells, human CTL1 was suggested to be an "intermediate-affinity" choline transporter, with a $K_m$ of ~10 μM, and showed reduced transport activity in buffers with acidic pH (pH 6.0–6.5) and 125 mM NaCl ([27]). In addition, uptake of radiolabeled ethanolamine into human fibroblast cells revealed "high-affinity" ($K_m$ ~67 μM) transport by CTL1 and "low-affinity" transport by CTL2 ($K_m$ of ~300 μM) ([31]). It was also shown that

CTL1 and CTL2 are both localized to the plasma membrane and mitochondrial membranes ([31]). On the other hand, *X. laevis* oocyte assays suggested CherI, the only described CTL protein in *A. thaliana*, to be an extremely high-affinity choline transporter ($K_m$ = 7–9 nM) and to increase choline uptake at low pH (pH 5.0) ([24]). *In planta* mutations of the *cherI* gene resulted in clear phenotypes (e.g., reduced phospho- and choline levels in fully developed leaves and stunted growth), but supplying *cherI* mutant plants with exogenous choline did not rescue these strong phenotypes ([28]).

In the current study, pH dependency assays of CTL1, PNS1, and CherI were conducted with 0.5 and 2 μM choline, whereas initial activity tests were done at 100 μM (Fig S1E and F). None of these proteins displayed activity above the level of the negative control for any of these conditions in our hands, even using 100 μM choline.

PNS1 has no previous reports of transport activity, whereas CherI has been reported to have a $K_m$ in the low nM range, and would have been expected to display activity. For CTL1, the pH assays are done at concentrations matching the first reported $K_m$ (1 μM for tCTL1) but below the later reported $K_m$ for human CTL1 of 10 μM ([22], [27]). However, as no transport was observed using 100 μM choline at pH 7.4, in contrast to what was reported by Inazu et al ([27]) (Fig 2A), we did not conduct additional pH dependency assays for any of the CTL proteins.

As CTL1, PNS1, and CherI did not show any activity for either choline or ethanolamine in standard ND96 buffer with 96 mM NaCl, further sodium dependency assays were also not done.

Under our assay conditions, only FLVCR1 and FLVCR2 showed significant transport of choline and ethanolamine (Fig 2A and B), leading to the speculation that CTL proteins might not directly transport these compounds. This has also been reported and speculated by others ([25], [33]). In particular, our results coincide with those of Nguyen and coworkers, who could not observe choline import by CTL1 or CTL2 in HEK cells ([33]).

The structure of CTL1 was the first of the SLC44 family to be determined and reveal a unique and, for a transporter, unusual arrangement of the 10 transmembrane helices ([17]). When superposing this model of CTL1 with the model we obtained, the models are almost identical ($RMSD_{Ca}$ = 0.8 Å), with certain regions better defined in the one structure than the other. However, the extracellular domain and most of M3/M4 are not possible to build in either structure (Fig S5B). One notable difference is that the previously determined CTL1 structure was a dimer, whereas we clearly observe a monomeric state in both SDS–PAGE gels, SEC profile, and micrographs. It remains unclear whether a dimeric or monomeric form is the physiological state of CTL1 and experimental protocol differences might explain the discrepancy.

CTL1 has a predicted molecular weight of 73 kD. The HEK cell purification that yielded the first structure of CTL1 shows an SDS–PAGE gel with multiple bands between 55 and 72 kD, similar to our purification from insect cells, but with some variations in migration pattern that can be attributed to different gel

---

sterols likely bind, whereas the putative binding pocket mostly contains hydrophilic residues. **(F)** Electrostatic coloring of PNS1 shown from the front, back, and top. The outer surface of PNS1 is mostly neutral to positively charged, whereas distinct positive (Arg296, Arg354, Lys389, and Arg495) and negative (Glu293, Glu393, and Glu492) patches are observed within the proposed binding pocket.

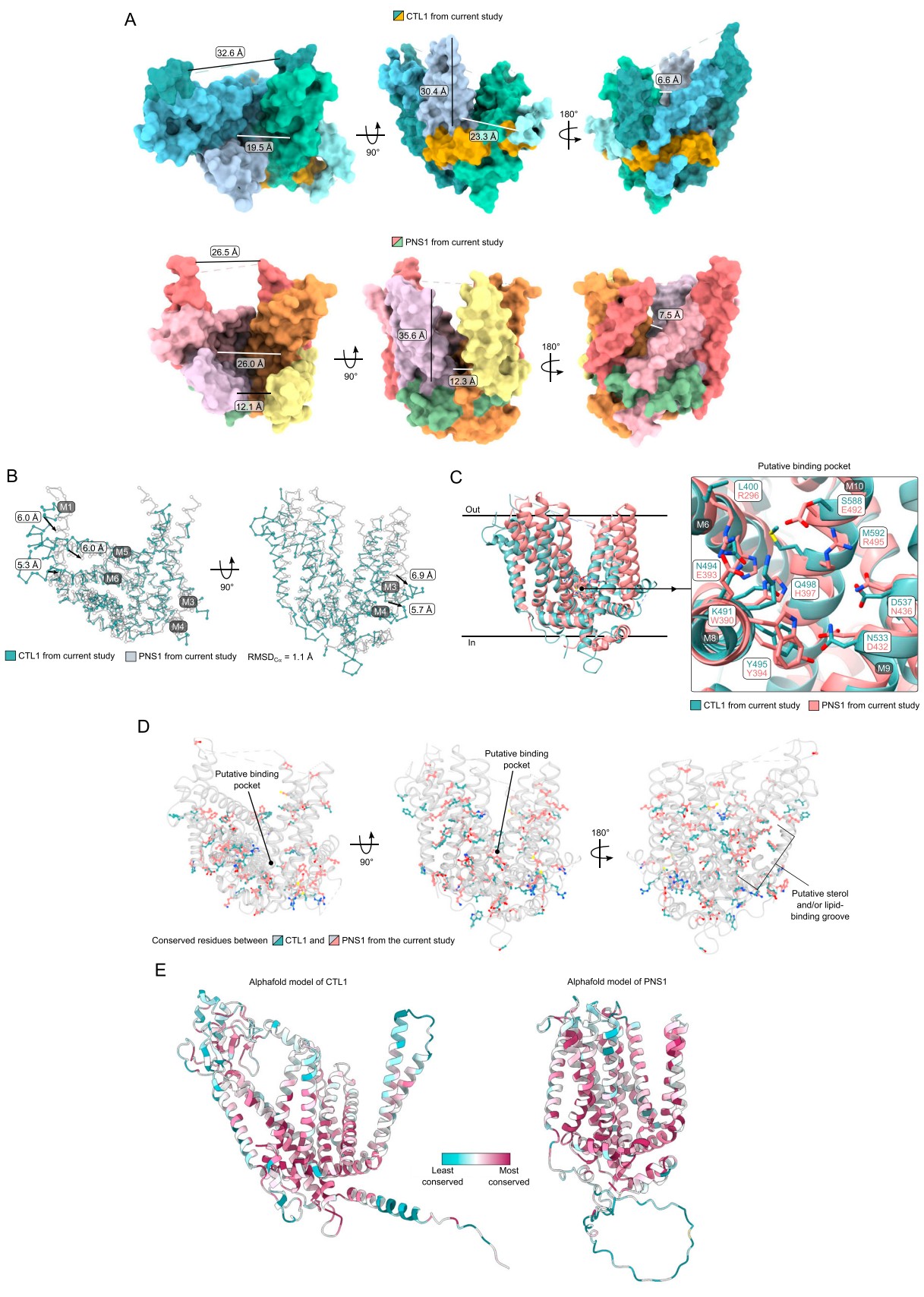

percentages and expression systems used (17). Nevertheless, SDS–PAGE and SEC profiles could raise questions about the integrity of the purified samples. In both cases, we use the sample only to obtain cryo-EM maps. The cryo-EM maps from our study and previously published studies match with full-length CTL1 (17). Even though the mobile regions were not built in either model, they are visible in the maps, indicating intact protein in both instances. Also, both models superpose well with AlphaFold predictions. These arguments support that our and previous published structures are representative of human CTL1.

We determined the structure of yeast PNS1 by crystallography with experimental phasing to verify the unusual fold of human CTL1 and confirm that it is representative of CTL family members. Notably, PNS1 does not contain the extracellular domain found in CTL1, but it does adopt the same overall conformation and is also in a monomeric state. Although the structure of PNS1 is more closed-off compared with that of CTL1, and the cavity of PNS1 seems broader and deeper compared with that of CTL1, it is clear that the overall structure is similar (Fig 6A).

Lastly, we searched for proteins with folds similar to CTL1 and PNS1 to gain insight into possible alternative functions of CTL proteins because not many residues are conserved between the two proteins (Fig S6). Based on structural similarity to PNS1, CysZ, a prokaryotic membrane permease of sulfate ($SO_4^{-2}$) used in cystine biosynthesis, was an unexpected result (Fig S7A). CysZ is proposed to adopt a hexameric arrangement as a trimer of dimers with each monomer having six helices that span the membrane to differing extents (41). Their proposed membrane topology thus differs significantly from the topology and oligomerization state we observe for PNS1 and CTL1, and it is hard to reconcile with any functional similarity. SLC25 members also featured on the list of structurally similar proteins to both CTL1 and PNS1, but with quite weak scores. SLC25 proteins contain six transmembrane helices organized in a barrel-like structure, surrounded by three amphipathic helices at the inner membrane plane toward the mitochondrial matrix (42, 43) (Fig S7B). Members of the SLC25 family localize specifically to mitochondria and generally function as exchangers, with SLC25A48 (Fig S7C) being an established choline transporter (10). Despite some similarities with these other structures, CTL1 and PNS1 still have an unusual shape and it is difficult to envision how it could facilitate an alternating access mechanism of transport (44), as proposed for members of the SLC25 family (43), for example.

The structural prediction of CherI looks like a combination of CTL1 and PNS1 because it is predicted to have an extracellular domain like CTL1 but is overall more closed-off like PNS1 (Fig S7C). One speculative function of CTL proteins could be regulation of protein sorting from the *trans*-Golgi network to the plasma membrane by way of lipid homeostasis because CherI mutants in

*Arabidopsis* were linked to altered levels of phospho- and sphingolipids (45).

In conclusion, by combining data from oocyte uptake assays with structures from cryo-EM and X-ray crystallography, we confirm and expand on the role of FLVCR proteins in the transport of choline and ethanolamine. Our data show that CTL proteins CTL1, PNS1, and CherI do not transport choline in oocytes under the conditions tested, but it cannot be discounted that these proteins might have some transport functionality under different experimental conditions. The weak structural similarity of CTL proteins to SLC25 could suggest a role as exchangers, but their closed bowl-like shape does not seem to support any known transport mechanism. Thus, we speculate that CTL proteins could possibly be involved in unidentified regulatory functions related to choline metabolism.

# Materials and Methods

## *X. laevis* oocyte assays

The cDNA from either *H. sapiens flvcr1* (UniProt: Q9Y5Y0), *H. sapiens flvcr2* (UniProt: Q9UPI3), *H. sapiens ctl1* (UniProt: Q8WWI5), *A. thaliana cheri* (UniProt: Q94AN2), or *S. cerevisiae pns1* (UniProt: Q12412) was cloned into the pNB1-U vector comprising a C-terminal GFP fusion (38).

For cDNA preparation, the genes were amplified by PCR using standard primers with 5' and 3' untranslated regions (forward, TTAACCCTCACTAAAGGGTTGTAATACGACTCACTATAGGG; reverse, TTTTTTTTTTTTTTTTTTTTTTTTTTTTTTTTTATACTCAAGCTAGCCTCGAG) followed by PCR purification (GeneJET PCR Purification; Thermo Fisher Scientific). RNA was synthesized by in vitro transcription using the mMESSAGE mMACHINE T7 Transcription Kit (Thermo Fisher Scientific). *X. laevis* oocytes were purchased from EcoCyte Bioscience in accordance with relevant guidelines and regulations. Capillary needles for RNA injection were made using a PC-10 micropipette puller (Narishige), and each oocyte was microinjected with 25 ng of RNA using the Nanoject III (Drummond Scientific), followed by incubation at 16°C for 3 d in Modified Barth's Solution (EcoCyte Bioscience) with 10 IU ml/ml gentamicin (Invitrogen).

The radioactive compounds were ordered from ARC, and all experiments were performed in ND96 buffer (96 mM NaCl, 2 mM KCl, 1.8 mM $CaCl_2$, 1 mM $MgCl_2$, and 5 mM Hepes-NaOH, pH 7.4) at pH 7.4, except for the pH dependency assay at pH 5.5, 6.0, 7.0, 7.4, or 8.5. Choline uptake assays were performed at concentrations ranging from 1 $\mu$M to 1 mM choline chloride (Sigma-Aldrich) using [$C^{14}$] choline chloride with 0.5–1 $\mu$Ci/ml. Uptake assays with [$H^3$] choline, [$H^3$] ethanolamine, [$H^3$] carnitine, and [$H^3$] histamine were done at

**Figure 6. Comparison of SLC44 structures determined in the current study.**
**(A)** Depth and width of the bowl-like cavity of CTL1 and PNS1 shown from the top, front, and back. **(B)** Superposition of CTL1 and PNS1 reveal that transmembrane M3/M4 of CTL1 is arranged in a manner that makes its structure more open compared with that of PNS1. Helices M1, M5, and M7 are also more open in CTL1 compared with PNS1. **(C)** Putative binding pockets of CTL1 and PNS1 are very different because only Y495$_{CTL1}$ and Y394$_{PNS1}$ are conserved. **(D)** Superposition of CTL1 and PNS1 with cartoons in gray and conserved residues, based on the sequence alignment of Fig S6, shown as sticks (teal for CTL1 and coral for PNS1). The models are superposed and shown from the top, front, and back. **(E)** AlphaFold-predicted models of full-length CTL1 (left) and PNS1 (right) colored according to evolutionary conservation of residues by ConSurf.

radioactive concentrations of 1–10 µCi/ml, using 100 µM substrate concentrations.

For the uptake assay, oocytes were incubated in ND96 buffer at a given substrate concentration for 10–60 min. The reaction was stopped by removal of the radioactive buffer and three consecutive washing steps in Modified Barth's Solution buffer. Oocytes injected with pure water instead of RNA, which were incubated in the same way as oocytes with RNA, served as negative controls.

The oocytes were disrupted by the addition of 100 µl of a 10% SDS solution while sonicating for 5 min in a water bath at 45°C. A volume of 3 ml OptiPhase HiSafe 3 scintillation fluid (PerkinElmer) was added to each vial, and radioactivity was quantified using Tri-Carb 4910TR Liquid Scintillation Counter (PerkinElmer). Experiments were performed at least in triplicate.

## Confocal microscopy of *X. laevis* oocytes

To confirm the expression of the target proteins by the oocytes, we looked for GFP signal with a confocal microscope. Images were taken using a Zeiss LSM 780 Axio Imager 2 confocal laser scanning microscope with the operating software ZEN v.3.5. Images were taken using a 10× objective with an excitation wavelength of 488 nm using 2.55% laser power with a frame time of 7.75 s, displaying a quarter of an optical slice of a 3-d post-injection oocyte.

## Cultivation of yeast cells to express FLVCR2 or PNS1

The gene encoding *H. sapiens* FLVCR2 (UniProt: Q9UPI3) was commercially sequence-optimized (for yeast expression) and synthesized by Twist Bioscience. This construct contains an N-terminal fiducial marker of three Protein A repeats (37), which replace the first N-terminal 82 amino acids of the FLVCR2 protein. The gene fragment was cloned into an expression vector based on p423_GAL1, comprising a C-terminal thrombin cleavage site and a 10×His-Tag (46). The FLVCR2-encoding plasmid was transformed into the *S. cerevisiae* expression strain DSY-5, grown in a selection media in shaker flasks, and harvested after a 22-h induction with galactose. Harvested cells were washed in cold water, centrifuged at 2,500*g*, at 4°C, for 10 min (8KS centrifuge with the 11,806 rotor; Sigma-Aldrich), and stored at –80°C.

The gene encoding *S. cerevisiae* PNS1 (UniProt: Q12412) was cloned into an expression vector based on p423_GAL1, comprising a C-terminal thrombin cleavage site and a 10×His-Tag (46). The PNS1-encoding plasmid was transformed into the *S. cerevisiae* DSY-5 cells and grown in a culture vessel to a high density by a fed-batch modality and harvested after a 22-h induction using galactose (47). Cells containing PNS1 protein were harvested as described above.

To prepare membranes containing either FLVCR2 or PNS1 protein, 300*g* of cells was thawed and resuspended in 750 ml lysis buffer (600 mM NaCl, 100 mM Tris–HCl, pH 7.5), supplemented with 1 mM PMSF, followed by 5 × 1 min cycles of bead beating with 0.5-mm glass beads (BioSpec Products). Lysed cells were centrifuged (9,000 rpm, 20 min, at 4°C in a Sorvall Lynx 6000 centrifuge, using the F9-6x1000 LEX rotor; Thermo Fisher Scientific), and the supernatant was subsequently ultracentrifuged at 42,000 rpm, at 4°C, for 2 h (Sorvall wX+ Ultra series using the T-647.5 rotor; Thermo

Fisher Scientific). Pelleted membranes were homogenized in membrane buffer (500 mM NaCl, 50 mM Tris–HCl, pH 7.5, 20% glycerol), flash-frozen in liquid nitrogen in aliquots of 3*g*, and stored at –20°C.

## Expression and membrane preparation of CTL1

The gene encoding *H. sapiens* CTL1 (UniProt: Q8WWI5) was commercially sequence-optimized (for yeast expression) and synthesized by GenScript. It was cloned into a modified pFL vector under the control of the very late polyhedrin promoter in frame with a C-terminal thrombin cleavage site and a triple Strep-Tag. The expression plasmid was transformed into *E. coli* (EMBacY cells) to integrate the DNA via Tn7 transposition into a baculovirus genome. The Tn7 transposition site was embedded in a lacZα gene allowing the selection for positive clones via blue/white screening using Bluo-Gal. Recombinant BACs were isolated via ethanol precipitation and used for transfection into ExpiSf9 cells (Invitrogen).

For initial virus (V0) production, the isolated BAC DNA was transfected into adhesive ExpiSf9 cells in six-well plates. Transfection efficiency was monitored via eYFP fluorescence. The initial virus was harvested ~60 h post-transfection and subsequently used for further virus amplification (V1) in a 25 ml suspension culture of ExpiSf9 cells. The amplified virus was harvested 60 h after cell proliferation was arrested. For large-scale expressions, cells were infected with the V1 virus at 1–2 × 10$^6$ cells/ml and harvested when the eYFP expression reached a plateau.

To prepare membranes containing CTL1, 100*g* of cells was thawed in 200 ml lysis buffer supplemented with protease inhibitor cocktail tablets from Thermo Fisher Scientific (1 tablet per 100 ml), Benzonase from Merck (1 mg/ml), and BioLock from Thermo Fisher Scientific (500 µl/100 ml). The cell suspension was divided into 2× 100 ml portions and sonicated with a Hielscher UP100H sonicator, using an MS7 sonotrode (70% intensity, 1 s on and 1 s off for 20 min). Lysed cells were centrifuged, and the supernatant was subsequently ultracentrifuged as described above. Pelleted membranes were homogenized in membrane buffer and stored as stated before.

## Purification of FLVCR2 for cryo-EM

Frozen membranes were thawed on ice and solubilized in basis buffer (200 mM NaCl, 50 mM Tris–HCl, pH 7.5, 10% glycerol, 5 mM choline chloride, 20 mM imidazole) containing 0.7% DDM and 0.07% CHS for 1 h at 4°C. Solubilized membranes were sonicated for 1 min, centrifuged at 15,000*g* for 20 min, and passed through a 1.2-µM pore filter (Acrodisc; PALL), before loading on a 1-ml Ni-NTA column (Cytiva). After washing with 20 mM imidazole in basis buffer for 8 ml and 70 mM imidazole in basis buffer for 15 ml, the protein was eluted in basis buffer supplemented with 400 mM imidazole. Cleavage of the purification tag was performed using thrombin (Avantor) overnight at 4°C while dialyzing into SEC buffer (50 mM NaCl, 20 mM Tris–HCl 7.5, 25 mM choline chloride, 0.4 mM TCEP, 0.02% DDM, 0.002% CHS). The next day, the sample was subjected to 0.5 ml Ni-NTA for removal of contaminants, as well as uncleaved protein, and the resin was washed with 2× 0.5 ml dialysis buffer. The

flow-through was concentrated using a 100 kD MWCO concentrator (Millipore, Sigma-Aldrich) and loaded on a S200 Superdex Increase 10/300 column (Cytiva) equilibrated in SEC buffer. The peak fraction was concentrated to 7.4 mg/ml and used for preparing grids for cryo-EM. For SDS–PAGE, samples were taken throughout the purification, added to loading dye, and loaded (without boiling) onto a 4–20% Bis-Tris mPAGE precast gel (Merck).

### Purification of CTL1 for cryo-EM

Membranes were thawed and resuspended in basis buffer supplemented with 5 mM choline chloride, 0.7% DDM, 0.07% CHS, and BioLock on a benchtop rotator at 4°C for 1 h. After sonicating 2 × 1 min at 70% for 1 s on and 1 s off (Hielsher UP100H sonicator, using an MS7 sonotrode), the debris was pelleted by centrifugation (9,000 rpm, 30 min, 4°C in a Sorvall Legend X1R centrifuge with a High Conic II SN1097 rotor; Thermo Fisher Scientific). The supernatant was filtered through a 1.2-$\mu$m syringe filter (Acrodisc; PALL Life Sciences), added to Strep XT beads (400°$\mu$l [as 800°$\mu$l 50% slurry]), and incubated at 4°C on a rotator for 2 h. The sample was centrifuged (300$g$ at 4°C for 5 min) to pellet the beads. The supernatant was added to beads (equilibrated with 200 mM NaCl, 50 mM Tris–HCl, pH 7.5, 10% glycerol, 5 mM choline chloride, 0.05% LMNG) in a 10-ml gravity column at 4°C and sequentially washed with 2× 1 ml of buffer containing 0.01% LMNG. The beads were washed with 3 ml G-buffer (150 mM NaCl, 20 mM Tris–HCl 7.5, 5 mM choline chloride, 0.005% LMNG). After adding 1 ml of G-buffer, the sample was transferred to a 2-ml Eppendorf tube containing thrombin, to cleave off the streptavidin tag, and incubated at 4°C while rotating overnight. The next day, the sample was washed with G-buffer and concentrated to 200 $\mu$l using a 0.5-ml Amicon centrifugal concentrator of 100 kD cutoff and injected onto an S200 Superdex Increase 10/300 (Cytiva) column pre-equilibrated with G-buffer. Peak fractions were pooled and concentrated to 5.5 mg/ml and used to prepare grids for cryo-EM. Before blotting some of the samples, 2 mM of HC-3 inhibitor was added to the protein, whereas the remaining samples were blotted without inhibitor.

Peptidisc peptides were synthesized by GenScript based on the description in reference (48). Purification of CTL1 in peptidisc was done in a similar manner to the sample in detergent, but, instead of exchanging from DDM to LMNG, the extra wash steps facilitated DDM-to-peptidisc exchange (48, 49, 50). Specifically, after, the protein was added to the slurry of Strep XT beads, in basis buffer containing 5 mM choline chloride, 0.1% DDM, 0.01% CHS, on a 10-ml gravity flow column at 4°C. The detergent and salt concentrations were lowered by washing with 2× 1 ml low-salt buffer (150 mM NaCl, 20 mM Tris–HCl 7.5, 5 mM choline chloride, 0.02% DDM, 0,002% CHS). The salt and detergent concentrations were lowered further by making a 2× dilution (75 mM NaCl, 20 mM Tris–HCl 7.5, 0.01% DDM, 0.001% CHS), while supplementing with choline chloride to keep the concentration 5 mM, and adding 8 ml to flow through the column. The exchange into peptidisc was done by adding 1.5 ml of peptidisc buffer (75 mM NaCl, 20 mM Tris–HCl 7.5, 5 mM choline chloride, 0.5 mg/ml peptidisc) to the column, letting the buffer flow through, reapplying the flow-through to the column, and repeating for 20 min. The beads were washed with 1 ml low-salt buffer,

transferred to a 2-ml tube, and supplemented with thrombin to cleave off the streptavidin tag and incubated at 4°C while rotating overnight. For gel filtration, the sample was processed in the same way as when in detergent. The final buffer contained 200 mM NaCl, 20 mM Tris–HCl 7.5, 5 mM choline chloride, and fractions were concentrated to ~5 mg/ml for grids; however, the true concentration of the protein was unclear because the peptidisc also contributes to the protein concentration measurement. For SDS–PAGE, samples were taken throughout the purification, added to loading dye, and loaded (without boiling) onto a 4–20% Bis-Tris mPAGE precast gel (Merck).

### Cryo-EM sample preparation and data collection for FLVCR2 and CTL1

QUANTIFOIL or C-flat R1.2/1.3 300 mesh grids were glow-discharged for 45 s at 15 mA in a GloQube Plus (Quorum). A 4 $\mu$l drop of sample was applied to the grid and blotted with a Vitrobot Mark IV (Thermo Fisher Scientific) operating at 4°C and 100% humidity, wait time of 10 s, blot time of 4 s, and blot force of 1, before plunge-freezing into liquid ethane.

A Titan Krios G3i microscope (Thermo Fisher Scientific), operating at 300 kV and equipped with a BioQuantum Imaging Filter (energy slit width of 20 eV) with a K3 detector (Gatan), was used to collect the movies. The datasets were acquired using automated acquisition EPU software at a nominal magnification of 1,300,00×, which corresponds to a physical pixel size 0.647 Å. For all datasets, the movies were saved in super-resolution pixel size and binned 2× in EPU back to the nominal pixel size.

### Processing and refinement of cryo-EM data for FLVCR2 and CTL1

Data were processed in cryoSPARC for both FLVCR2 (Fig S2) and CTL1 (Figs S3A and 3F) (51). Briefly, patch motion correction and CTF estimation were performed and low-quality micrographs were discarded before an initial set of particles was picked via the blob picker and extracted. Several rounds of 2D classification were performed with selected classes. After an initial volume was obtained from ab initio reconstruction, iterative rounds of hetero-refinement with "good" ab initio classes and "junk" classes were performed and fed into ab initio reconstruction to improve the volume for template picking. For FLVCR2, iterative rounds of hetero-refinement with two good ab initio volumes were used to separate good particles from junk. Later, nonuniform (NU) refinement maps were generated with improved particle stacks until there was no more improvement in the resolution. Local refinement gave the final map, from where the local resolution was estimated.

For CTL1, to try improving the quality of the map in the extracellular domain and M3/M4, 3D classification was done and the best class with the most particles was used for the final local refinement and local resolution estimation.

To build FLVCR2 and CTL1, models predicted by AlphaFold (52) were used. Initial fitting was done using Namdinator (53) and the ISOLDE plugin (54) of ChimeraX (55). Coot (56) was used to manually rebuild the models and correct outliers, whereas the models were refined in Phenix (57). The models were uploaded to the PDB (58),

whereas maps and data collection details were deposited to the EMDB (59).

### Purification of PNS1 for crystallography

PNS1 was purified in a similar manner to FLVCR2; however, some modifications were made to have a sample suitable for crystallography. For LCP crystallization, the protein loaded onto the Ni-NTA column was washed with 8 column volume (CV) basis buffer containing 0.1% DDM and 0.01% CHS and 15 CV basis buffer containing 50 mM imidazole and 0.1% DDM and 0.01% CHS. The protein was eluted in G-buffer LCP (200 mM NaCl, 20 mM Tris–HCl 7,5, 5% glycerol, 0.025% DDM, and 0.0025% CHS) containing 400 mM imidazole. The protein was cleaved by trypsin (1:250 w/w) overnight at 4°C. Trypsin was inhibited using trypsin inhibitor from soybean; the protein was concentrated (50 kD cutoff: Millipore, Sigma-Aldrich) and loaded on a S200 Superdex Increase 10/300 column (Cytiva) equilibrated in G-buffer LCP. Peak fractions were concentrated to 35 mg/ml and either used directly in crystallization trails or flash-frozen and stored at −80°C. For one half of the sample, we tested binding of HC-3 by the addition of 3 mM HC-3 before LCP setup.

For vapor diffusion crystallization, detergent was exchanged to OGNG on the Ni-NTA column. After washing with basis buffer as described before, the column was washed with 25 CV G-buffer OGNG (200 mM NaCl, 20 mM Tris–HCl 7.5, 5% glycerol, 0.12% OGNG) and subsequently eluted with G-buffer OGNG containing 400 mM imidazole. The protein was cleaved by trypsin (1:400 w/w) overnight at 4°C. Trypsin was inhibited using trypsin inhibitor from soybean; the protein was concentrated (50 kD cutoff; Millipore, Sigma-Aldrich) and loaded on an S200 Superdex Increase 10/300 column (Cytiva) equilibrated in G-buffer OGNG. The first peak was concentrated to 20 mg/ml and either used directly in crystallization trails or flash-frozen and stored at −80°C. For SDS–PAGE, samples were taken throughout the purification, added to loading dye, and loaded (without boiling) onto a 4–20% Bis-Tris mPAGE precast gel (Merck).

### Crystallization of PNS1

For crystallization in LCP, concentrated PNS1 was mixed with molten mono-olein in a 2:3 ratio using two Hamilton syringes and a coupler. PNS1 was crystallized in a 96-well format using glass sandwich plates and a Crystal Gryphon (ARI) using 50 nl LCP mix and 1 μl reservoir. Initial crystal conditions were identified using the commercially available "JBScreen LCP" crystallization screen and further optimized using grid screening to the final crystallization condition of 0.1 M $(NH_4)_2PO_4$, 0.1 M Hepes 7.0, 32% PEG 400, 6 mM TCEP. Crystals appeared after a few days and grew to full size (~50 μm) within 2 wk. Crystals were fished and flash-frozen in liquid nitrogen without further cryoprotection.

For experimental phasing, we identified hits with vapor diffusion crystallography conditions. Here, PNS1 was crystallized by mixing protein and reservoir (0.3 M sodium formate, 0.1 M sodium citrate buffer, pH 5.0, 15% PEG 3350, 0.7–1.5% 1-butanol) in a 1:1 or 1:1.5 ratio in sitting or hanging-drop plates. Crystals grow to a size of 100 μm within a week. For single-wavelength anomalous diffraction (SAD)

experiments, PNS1 crystals were then soaked in 20 mM $KAu(CN)_4$ dissolved in the reservoir overnight into the crystallization drop.

### X-ray crystallography data processing and structure determination of PNS1

Initial experimental phases were obtained using the Crank2 (60) pipeline in CCP4I2 (61). An initial model was generated with trRosetta modeling server and the backbone heavily modified manually (RMSD > 3 Å) (62). Initial fitting of the model to the map was done using Namdinator (53). Model building and refinement were then done iteratively in Coot (63) and Phenix.refine (57). The model and relevant data were uploaded to the Protein Data Bank (58).

### Bioinformatics tools and figure preparation

Structural predictions were obtained from the AlphaFold database (64) that is linked to UniProt (65) and from trRosetta (62). Predictions of fusion proteins, like Protein A and FLVCR2, were performed locally using AlphaFold2 (52, 66). For evolutionary conservation of protein sequences, the ConSurf server with default settings was used (67). DALI (68) was used to search the PDB and AlphaFold database for proteins based on structural similarities to CTL1 and PNS1. Hits were obtained for CysZ (Z score of 9.7) and also for several SLC25 members (Z score of 8.6–8.8). Graphical representations of protein structures were made in ChimeraX v1.71 (55). Graphs were created in GraphPad Prism 10, and simple linear regression fits were used for time-dependent uptake assays, whereas Michaelis–Menten least squares fits were applied to assays when substrates were titrated to obtain $K_m$ values. Sequence alignments were done with PROMALS3D (69) and annotated using ALINE (70).

## Data Availability

Atomic models have been deposited in the Protein Data Bank (PDB), and cryo-EM maps have been deposited in the Electron Microscopy Data Bank (EMDB). FLVCR2: PDB 9QU4 and EMDB EMD-53371. CTL1: PDB 9QU3 and EMDB EMD-50252. PNS1: PDB 9F63.

## Supplementary Information

## Acknowledgements

We acknowledge the EMBION Cryo-EM Facility at Aarhus University (application #0137). We also acknowledge eBIC for cryo-EM data collection (BAG BI27980). We acknowledge access to the computational infrastructure at the Center for Structural Biology. We thank beamlines I24 and I04 at the Diamond Light Source and beamline BioMAX at the MAX IV Laboratory, where X-ray data were collected, and DESY-PETRA III for crystal screening. Funding

was provided by a Lundbeckfonden fellowship (R380-2021-1207) and a Marie Skłodowska-Curie Actions fellowship (#890822) to JH Driller. This work was supported by the Danish Council for Independent Research (Grant Agreement No. 0135-00032B), the Novo Nordisk Foundation (grant agreement no. NNF24OC0088380) and the Carlsberg Foundation (Grant Agreement No. CF19-0127) to BP Pedersen. This work was supported by the Danish National Research Foundation under the grant DNRF190 – Center for active transport of plant hormones (Plant-PATH).

## Author Contributions

L Nel: data curation, formal analysis, validation, investigation, visualization, methodology, project administration, and writing—original draft, review, and editing.
JH Driller: conceptualization, data curation, formal analysis, validation, investigation, visualization, methodology, project administration, and writing—original draft, review, and editing.
R Driller: formal analysis, supervision, validation, methodology, and writing—original draft, review, and editing.
KM Frain: investigation, and writing—original draft, review, and editing.
BP Pedersen: conceptualization, resources, supervision, funding acquisition, and writing—original draft, review, and editing.

## Conflict of Interest Statement

The authors declare that they have no conflict of interest.

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
