## [Reviewer comments · Life Science Alliance]

Structural and biochemical comparison of the FLVCR and CTL membrane protein families in eukaryotes

Lynette Nel, Jan Driller, Ronja Driller, Kelly Frain, and Bjørn Pedersen

DOI: <https://doi.org/10.26508/lsa.202503583>

Corresponding author(s): *Bjørn Pedersen, Aarhus University*

Review Timeline:

Submission Date:	2025-11-26
Editorial Decision:	2026-01-06
Revision Received:	2026-02-24
Editorial Decision:	2026-03-30
Revision Received:	2026-04-23
Accepted:	2026-04-30

Scientific Editor: *Sarita Hebbar*

Transaction Report:

January 6, 2026

Re: Life Science Alliance manuscript #LSA-2025-03583

Dr. Bjørn Panyella Pedersen
Aarhus University
Department of Molecular Biology and Genetics
Universitetsbyen 81
MBG-AU
Aarhus C, Danmark 8000
Denmark

Dear Dr. Pedersen,

Thank you for submitting your manuscript entitled "Biochemical and structural comparison of eukaryotic choline transport protein families FLVCR and CTL" to Life Science Alliance.

Your manuscript was reviewed by three expert reviewers whose comments are appended below. As you will note, the reviewers comment differently on the solidity of the data and on required validation for conclusions. We concur with Reviewer 3 that although the work lacks mechanistic details, it is of potential interest to the field.

Reviewers 1 and 2 have raised concerns on the uptake assays, which we agree must be addressed. These include a comparison of expression levels to enable comparison of uptake (Reviewer 1, comment 2). We additionally concur with all the reviewers that functionality of the chimeric constructs must be demonstrated using any of the approaches suggested by Reviewer 1 (comment 6), or by Reviewer 2 (major point 1) and Reviewer 3 on pH-dependent transport assays.

Further in terms of protein preparation, we agree with the points of Reviewer 2 and 3 that the oligomerisation state of CTL1 must be specified with a SDS-PAGE gel and SEC trace shown in the workflow. Here we also point you to all comments of the three reviewers that all methodological details, rationale, separate workflow, with representative classes, # of particles, and images of maps, peptidisc preparation/incorporation should be provided. In this connection, we ask you to pay heed to the suggestion of Reviewer 1 on providing a proper rationale for Alpha-fold predictions or removing them from the manuscript.

Finally we concur with the reviewers that you must provide appropriate evidence or moderate your conclusions claiming in-vivo functionality (Reviewer 1, comment 1), robust transport in case of FLVCR1 (Reviewer 1 comment 3), and binding of Choline to CTL1 (Reviewer 3, comment paragraph 5). Additionally in agreement with the reviewers we ask that you reframe the conclusions in the context of presented results (density interpretation, by Reviewer 1, point 7 and CTL1 in MNG vs peptidisc, Reviewer 3, comment paragraph 1) and previous publications (Reviewer 2, major point 1).

We invite you to submit a revised manuscript addressing the reviewers' comments. When submitting the revision, please include a letter addressing the reviewers' comments point by point. While a rebuttal must respond to all points in some form, additional experiments to resolve these points, other than indicated above, are not required.

I would be happy to discuss the revision in more detail via email or phone/videoconferencing. Please let me know which option you prefer, if any.

Thank you for this interesting contribution to Life Science Alliance. We hope that the comments below will prove constructive as your work progresses, and we are looking forward to receiving your revised manuscript.

Sincerely,

- A letter addressing the reviewers' comments point by point.
- An editable version of the final text (.DOC or .DOCX) is needed for copyediting (no PDFs).
- High-resolution figure, supplementary figure and video files uploaded as individual files: See our detailed guidelines for preparing your production-ready images, <https://www.life-science-alliance.org/authors>
- Summary blurb (enter in submission system): A short text summarizing in a single sentence the study (max. 200 characters including spaces). This text is used in conjunction with the titles of papers, hence should be informative and complementary to the title and running title. It should describe the context and significance of the findings for a general readership; it should be written in the present tense and refer to the work in the third person. Author names should not be mentioned.
- By submitting a revision, you attest that you are aware of our payment policies found here: <https://www.life-science-alliance.org/copyright-license-fee>

B. MANUSCRIPT ORGANIZATION AND FORMATTING:

Reviewer #1 (Comments to the Authors (Required)):

In this report by Nel, Driller, Driller and colleagues, the authors investigate the molecular basis of choline uptake into mammalian cells by FLVCR1, FLVCR2 and CTL1. Structures of FLVCR2 in the presence of choline, CTL1 in an unknown state and PNS1 in an unknown state are reported. Functional analyses of FLVCR1/2 family members and CTL1 families are also reported. These results are consistent with recent structural and functional analyses of both families of membrane transport proteins. While the structural data adds to the understanding of choline transport, the *Xenopus laevis* oocyte uptake experiments lack the rigor necessary to support several of the authors' key conclusions. In their current form, the transport data do not convincingly demonstrate robust choline or ethanolamine transport by FLVCR proteins, nor do they provide sufficient evidence to conclude that CTL proteins lack such activity. Additional controls, expression quantification, and consistent presentation of negative controls and substrate comparisons are required. Several figures also require clearer presentation, and some structural interpretations are overstated or not sufficiently supported by the data. These issues should be addressed prior to publication.

Comments.

1. The abstract refers to "in vivo" transport activity; however, the functional data are solely derived from heterologous overexpression in *Xenopus* oocytes. This system is not sufficient to justify an "in vivo" claim and the wording should be revised accordingly.
2. The authors demonstrate that the GFP-tagged constructs are expressed in the plasma membrane of oocytes. However, lacking a quantification of their expression levels, it is difficult to compare the uptake between constructs. The authors should perform western blots to evaluate the expression levels across constructs. Without this, differences in uptake signals may reflect variable protein abundance.
3. The H₂O-injected oocyte controls show high uptake signals, in some cases comparable to those of FLVCR1 or FLVCR2. This suggests either substantial endogenous uptake or assay background. Importantly, the negative control signal does not differ from FLVCR1 in several panels (for example in S1D), contradicting the claim on line 104 of robust transport. The authors should revisit the results of their transport assays and revise their conclusions accordingly.
4. The concentration-response data for FLVCR1/2 presented in Figure S1C-D appear to be fitted with linear functions. The manuscript does not state what fitting functions were used, how parameters were derived, or whether nonlinear models were

tested. These methodological details should be added to the methods and the figure.

5. There are several inconsistencies in how the transport assays are presented. For example, time-dependent uptake is shown only for choline, while ethanolamine data are missing. Conversely, ethanolamine titration is shown, but choline titration is omitted. A consistent presentation of the analyses would help the reader to better interpret the results.

6. Several of the structures are determined using chimeric constructs. These constructs should be characterized using the radiolabeled transport assays to ensure that they represent functional transporters.

7. Unassigned densities are observed at low contour levels in the CTL1 and PNS1 maps. It is unclear if these correspond to bound ligands, water molecules or noise. The authors should be conservative in their interpretations of these densities.

8. The authors compare the density maps to predicted alphafold structures in several figures (3C and 4C). As there are previously determined experimental structures of FLVCR2 and CTL1, the values of these comparisons is unclear. The authors should better explain the rationale for these comparisons or remove them.

Minor comments:

1. Figure 2C combines FLVCR1 and FLVCR2 despite large differences in signal magnitude, compressing data and obscuring trends. It would help the reader to interpret these results if the data were separated into independent panels. Overall, several of the figures are small and overly compressed, making interpretation difficult; resizing is recommended.

2. The manuscript highlights putative sterol or lipid densities but provides no biochemical data to support their identity. Given the low contouring and uncertainty, these assignments should be toned down or removed.

3. A workflow for the image processing of CTL1 should be included.

4. Several of the figures employ highly complex color schemes, which makes it difficult to interpret the superpositions. For example, the features in Figure 6B-C are hard to distinguish.

Reviewer #2 (Comments to the Authors (Required)):

In this manuscript, Nel et al. perform functional characterization of FLVCR1/FLVCR2/CTL1/PNS1/Cher1. They also determine the structures of FLVCR2, CTL1 and PNS1. Although generally scientifically sound, the manuscript does not present substantial new insights beyond what is already established in the literature. In instances where its findings diverge from published work, the experiments do not appear to have been conducted under conditions appropriate for a valid comparison.

Major points

- The manuscript does not examine choline transport across a range of pH conditions for CTL1. Given that CTL1 is a proton-coupled antiporter, omission of pH-dependent transport assays represents a substantial gap. As currently presented, the functional data cannot be considered conclusive and would explain why this observation differs from the majority of papers in the literature.
- The FLVCR2 structure does not provide additional insight and is lower in resolution than previously published FLVCR2 structures. The rationale for including this dataset, and its incremental value to the field, requires clarification.
- No biochemical characterization/ purification figure of CTL1 is provided.
- The CTL1 structure, despite its marginally improved resolution, does not appear to yield new mechanistic insight compared with the previously published model. While they identify the presence of a density that appears as though it may not be choline, this density has not been characterized and therefore this does not provide any novel information.
- The FLVCR2 structure does not appear to provide additional mechanistic insight and is lower in resolution than previously published FLVCR2 structures. The rationale for including this dataset, and its incremental value to the field, requires clarification.
- The oligomeric state of CTL1 remains unclear. Prior analyses have suggested it exists as a dimer. The authors should comment on whether their construct forms a monomer or higher-order assembly in solution or in the membrane mimic.
- Method details for peptidisc preparation/incorporation are missing. Without these, reproducibility and interpretation of the structural data are limited.
- The title seems unusual seeing as though the authors are claiming that 2/3 or the transporters studied are not actually choline transporters.

Reviewer #3 (Comments to the Authors (Required)):

The manuscript entitled "Biochemical and structural comparison of eukaryotic choline transport protein families FLVCR and CTL" by Nel et al is a careful examination of choline transport (or lack thereof) by different transmembrane proteins. A major conclusion of this paper is that contrary to previous reports, CTL1-type proteins do not function as choline transporters, at least in the experimental conditions tested here. Impressively, the authors also report structures of human CTL1, FLVCR2 (cryo-EM) and the yeast CTL1 homologue, PNS1 (crystallography). The structure of human CTL1 was reported previously, in a dimeric

configuration, whereas here the authors show a monomer at an improved resolution. The structure of FLVCR2 in this conformation has also been reported before, though this structure here demonstrates the use of Protein A as an inbuilt fiducial, and fits well with the logical progression of the paper. The unusual fold of human CTL1 observed is verified by the crystal structure determination of a homologous protein, PNS1, from yeast. This supports that the quite fascinating open-to-the-membrane, bowl-like structure is conserved in this family. Although the authors are yet to uncover the mechanistic details of CTL1 function, I think this work a valuable contribution to the field, and the paper to be well-written & presented.

I do have several comments about the structural analysis of CTL1, that I think should be addressed before publication:

Regarding CTL1 in MNG vs peptidisc, I am not sure that there are no "observed differences between the two maps" (Line 151)- when I aligned the maps EMDB-50251 and EMDB-50252 it seems they align well on parts of the N-terminal portion of the transmembrane protein, but the M3/M4 helices in the peptidisc structure seem to have been pulled inward closer to the N-terminal portion of the protein. Could this possibly be a peptidisc mediated artefact? Could the authors clarify this, as although these differences are only visible at lower contours, they are still apparent in the maps. An overlay figure could be good here.

In the deposited model the authors chose not to model the extracellular domain, or the M3 and M4 helices. While I understand that the resolution is lower here, from inspection of the maps, it seems that placement of the extracellular domain is feasible. This could be achieved with the AlphaFold model and Isolde. I think that it would increase the utility of the structure. The authors might consider including this domain (and perhaps m3/m4) in the PDB deposition, perhaps with appropriately conservative modelling (truncate sidechains). DeepEMhancer or the EMReady2 web server might be used to aid modelling.

Did the authors try focused refinements with a mask around the N-terminal & extracellular domain, and/or 3DVA? Further to this- there is no workflow for CTL1. Even if similar to FLVCR2, the authors really should include a separate workflow, with representative classes, # of particles, and images of maps. I also think an SDS-PAGE gel should be shown with the workflow, and a SEC trace. Was there any hints of a dimer? It seems that PNS1 might have a monomer-dimer self-association per the gel and SEC trace in the Supplementary. I note again the previous structure of CTL1 was a dimer- I wonder if this is functionally relevant.

The functional characterisation in oocytes is convincing, with the uptake assay data in Figure 2 clear and presented very nicely (albeit a little small and hard to read, this should be addressed in the final version prior to publication). These data are strong evidence against CTL1 being a choline transporter, although, I wondered why pH/Na⁺ dependence of choline/ethanolamine uptake was not assayed for CTL1? It does seem unlikely that these are coupled symporters/antiporters given the architecture, however, given claims by Taylor et al, JBC (2021), that CTL1 functions as a Na⁺-independent, H⁺/ethanolamine antiporter, it seems that this should be tested, or at least be caveated in the discussion.

"Our data suggests that CTL proteins like PNS1, CTL1 and Cher1 do not bind choline and are not choline transporters." While I agree that the authors have shown these proteins are unlikely to be choline transporters, is it not possible that choline or choline containing molecules may bind but not be transported? This could be toned down. Further to this, if the authors are suggesting CTL proteins are not transporters, perhaps they should consider removing "choline transport" from the title.

Line 541: "mano-olin"-should this be mono-olein?

Reviewers' Comments

(authors reply in blue)

Reviewer #1

In this report by Nel, Driller, Driller and colleagues, the authors investigate the molecular basis of choline uptake into mammalian cells by FLVCR1, FLVCR2 and CTL1. Structures of FLVCR2 in the presence of choline, CTL1 in an unknown state and PNS1 in an unknown state are reported. Functional analyses of FLVCR1/2 family members and CTL1 families are also reported. These results are consistent with recent structural and functional analyses of both families of membrane transport proteins. While the structural data adds to the understanding of choline transport, the *Xenopus laevis* oocyte uptake experiments lack the rigor necessary to support several of the authors' key conclusions. In their current form, the transport data do not convincingly demonstrate robust choline or ethanolamine transport by FLVCR proteins, nor do they provide sufficient evidence to conclude that CTL proteins lack such activity. Additional controls, expression quantification, and consistent presentation of negative controls and substrate comparisons are required. Several figures also require clearer presentation, and some structural interpretations are overstated or not sufficiently supported by the data. These issues should be addressed prior to publication.

We thank the reviewer for the careful review of our manuscript. We find it has significantly benefited from the suggestions and comments made.

Comments.

1. The abstract refers to "in vivo" transport activity; however, the functional data are solely derived from heterologous overexpression in *Xenopus* oocytes. This system is not sufficient to justify an "in vivo" claim and the wording should be revised accordingly.

It is correct that there are very different traditions in various areas of molecular biology for the definition of "in vivo". For instance, in neuroscience, *in vivo* would imply an animal (e.g. mouse) was used, while in biophysics *in vivo* would be applied to any "living cell" system, including cell suspensions. This variation introduces ambiguity in interpretation, which should be avoided. To address this, we have changed the wording from "*in vivo*" to "*Xenopus laevis* oocytes" and adjusted the text any other places where this might cause confusion.

2. The authors demonstrate that the GFP-tagged constructs are expressed in the plasma membrane of oocytes. However, lacking a quantification of their expression levels, it is difficult to compare the uptake between constructs. The authors should perform western blots to evaluate the expression levels across constructs. Without this, differences in uptake signals may reflect variable protein abundance.

We appreciate the reviewer's comment, which points to a discussion that is often raised when doing oocyte assays. Both GFP fluorescence and Western Blot (WB) are at best semi-quantitative, and both fluorescence and WB are routinely used to (semi) quantify expression levels.

Values from WB or fluorescence are normally not used to derive actual V_{max} values due to the semi-quantitative nature of both methods. GFP fluorescence is often considered superior to whole-oocyte WB, as it directly targets the plasma membrane, while WB is done on whole oocyte lysate and not on isolated plasma membranes. WB can be done on isolated plasma membranes, but this adds even more steps that reduce any attempt at accuracy.

In our case, the fluorescence images were recorded with identical settings on the microscope to make comparisons meaningful. We therefore find that using GFP fluorescence to support expression levels is at par or even superior to WB.

A few examples of the varied standards for oocyte quantification are provided here:

- The O'Regan and Meunier 2003 paper (ref. 29), which studied CTL from *Torpedo marmorata* in *Xenopus* oocytes, did not image or quantify the expression levels of recombinant protein. They used radiolabeled choline uptake in the presence and absence of an inhibitor and various salts to evaluate choline transport (Fig. 3A-B).

- Similarly, no images of *Xenopus* oocytes nor quantification of fluorescent signal were shown for Cher1 by Dettmer et al., 2014 (ref. 24). The oocyte assays were used in time-dependent (Fig. 2E), choline titration (Fig. 2F), choline accumulation (Fig. S3A-B) and pH-dependency (Fig. S3C) experiments.

Other examples, not related to choline:

- A study by Zhang and co-workers (2025; doi.org/10.1016/j.jplph.2025.154576) showed that the sugar transporter *GmSWEET48* from soybean regulates fructose and glucose levels in seeds. They used *Xenopus* oocytes to compare sugar transport by *GmSWEET48* with that of *Arabidopsis* SWEET12 (Fig. 2B-C). No images or quantification of heterologous protein expression were provided.
- In a study by Nielsen et al., (2006; doi.org/10.1016/j.cub.2006.05.043), the auxin-influx carrier (AUX1) from *Arabidopsis thaliana* was characterized by using *Xenopus* oocytes, radioactive auxin and inhibitors. Confocal microscopy images (Fig. 1A) of water-injected and AUX1-YFP-injected oocytes were provided to show expression and membrane localization, without quantification, to support the uptake assays that followed.
- The monosaccharide transporter, STP13 (Figure 1 and Table 2), from *Arabidopsis thaliana*, was characterized by using *Xenopus* oocytes (Halkier, 2006, doi:10.1016/j.febslet.2006.03.064). No microscopy images to show expression of STP13 in oocytes were provided.
- Newstead and co-workers (2022, doi.org/10.1038/s41467-022-32589-2) studied cystinosin, a lysosomal cystine transporter found in humans and *Arabidopsis thaliana*. This study used two electrode voltage clamp of *Xenopus* oocytes to characterize cystinosin (Figure 4C) and confirmed expression of wild-type and mutant proteins with C-terminally GFP tagged constructs with confocal microscopy (Figure S6). No quantification of protein expression was performed.

3. The H₂O-injected oocyte controls show high uptake signals, in some cases comparable to those of FLVCR1 or FLVCR2. This suggests either substantial endogenous uptake or assay background. Importantly, the negative control signal does not differ from FLVCR1 in several panels (for example in S1D), contradicting the claim on line 104 of robust transport. The authors should revisit the results of their transport assays and revise their conclusions accordingly.

We agree that transport of FLVCR1, while clearly present, is not ‘robust’. We have toned down our conclusions and descriptions accordingly.

4. The concentration-response data for FLVCR1/2 presented in Figure S1C-D appear to be fitted with linear functions. The manuscript does not state what fitting functions were used, how parameters were derived, or whether nonlinear models were tested. These methodological details should be added to the methods and the figure.

We have added these details to both figures and methods. In short:

For time-dependent uptake assays (Fig S1C), a simple linear regression (SLR) was used.

For FLVCR1 and FLVCR2 with choline (Fig 2C, 2D) a Michaelis-Menten least squares fit was used.

For FLVCR1 and FLVCR2 with ethanolamine, saturation could not be obtained in our experimental setup. To emphasize this, we apply a Michaelis-Menten least squares fit, which is not able to calculate a K_m-value. We appreciate that in Fig S1D it looks like a simple linear regression, but it is not.

We now explicitly state this in the figure legend:

“(D) Titration curve of FLVCR1 and FLVCR2 with H³-ethanolamine. Saturation was not achieved and a K_m-value could not be determined when applying a Michaelis-Menten least squares fit. The plots appear to be fitted with a simple linear regression fit, but is in fact a Michaelis-Menten least squares fit.”

We provide a screenshot from GraphPad Prism to further support this:

Nonlin fit		A	B	C	D	E	F	G
Table of results		FLVCR1	FLVCR2	H2o				
1	Michaelis-Menten							
2	Best-fit values							
3	Vmax	3.809e+017	4.245e+017	5.885e+017				
4	Km	Unstable	Unstable	Unstable				
5	95% CI (profile likelihood)							
6	Vmax	??? to +infinity	??? to +infinity	??? to +infinity				
7	Km	(Very wide)	(Very wide)	(Very wide)				
8	Goodness of Fit							
9	Degrees of Freedom	32	31	25				
10	R squared	0.8640	0.9657	0.8712				
11	Sum of Squares	119140	64380	80983				
12	Sy.x	61.02	45.57	56.91				
13	Constraints							
14	Km	Km > 0	Km > 0	Km > 0				
15								
16	Number of points							
17	# of X values	42	42	42				
18	# Y values analyzed	34	33	27				
19								
20								

Michaelis-Menten
Least squares fit
Prism has identified at least one unstable parameter. This suggests that your data may be incomplete or don't fully describe the selected model. Because of this, confidence intervals for all parameters should be interpreted with caution.
Learn more
Equation help

5. There are several inconsistencies in how the transport assays are presented. For example, time-dependent uptake is shown only for choline, while ethanolamine data are missing. Conversely, ethanolamine titration is shown, but choline titration is omitted. A consistent presentation of the analyses would help the reader to better interpret the results.

We agree that a consistent presentation helps the narrative.

The titration curves of choline for FLVCR1 and FLVCR2 are shown in Figure 2C, 2D, while the titration curves (with no saturation observed) of ethanolamine are shown in Figure S1D. We would prefer to keep the ethanolamine titration curve as a supplementary figure since no saturation could be obtained.

6. Several of the structures are determined using chimeric constructs. These constructs should be characterized using the radiolabeled transport assays to ensure that they represent functional transporters.

In the manuscript we present 3 structures: FLVCR2, CTL1 and PNS1.

The structures of CTL1 and PNS1 are not from chimeric constructs. They contain a C-terminal affinity tag, but these are removed prior to structure determination.

The structure of FLVCR2 is from a chimeric construct with a 'Protein A' triple-repeat fused to the N-terminal end. We compare the obtained inward-facing conformation with published structures of FLVCR2 in the same conformation. These other structures come from non-chimeric and FAB-bound FLVCR2. As also discussed in the main text, the RMSD_{Ca} between these models and our model is < 1 Å (Figure S5A). Thus, there is no indication that the FLVCR2 structure we have obtained should represent a non-functional transporter. To further support that the FLVCR2 structure represents a functional transporter, we highlight that the oocyte transport assays were done with full-length FLVCR2, indeed without Protein A, but then instead fused to a C-terminal GFP. Expression and choline transport were observed for this chimeric FLVCR2-GFP, which indicates that FLVCR2 with a domain fused to the C-terminus is capable of transitioning between inward- and outward-facing conformations and constitutes a functional transporter.

7. Unassigned densities are observed at low contour levels in the CTL1 and PNS1 maps. It is unclear if these correspond to bound ligands, water molecules or noise. The authors should be conservative in their interpretations of these densities.

We agree that there are many unexplained map features in the CTL1 and PNS1 maps. At the resolutions we are working with, this is to be expected. We try to be very conservative in our interpretation. e.g. we have refrained from building parts of our protein models if we were not confident about the main chain trace. We keep these densities unassigned for exactly this reason as noted by the reviewer.

Some densities in the PNS1 map, however are of interest and we have shown them in Figure 5D. We clarify in the Figure 5 legend that the purple mesh indicates non-proteinaceous, but unassigned, densities.

8. The authors compare the density maps to predicted alphafold structures in several figures (3C and 4C). As there are previously determined experimental structures of FLVCR2 and CTL1, the values of these comparisons is unclear. The authors should better explain the rationale for these comparisons or remove them.

Figure 3C: For FLVCR2, as discussed in the main text, the map quality does not allow us to confidently model the complete Protein A repeat that was used as a fiducial marker. We have added the AlphaFold model to show where the Protein A segment of the chimeric construct would be and highlight that there is a clear map feature in that region, used for particle alignment.

Figure 4C: For CTL1, as discussed in the main text, the map quality does not allow us to confidently model the extracellular domain and the loop connecting M3 to M4. We have added the AlphaFold model to highlight which regions of the deposited CTL1 model are missing, and to show that these missing regions do have map features in the right position, that however cannot be interpreted.

For these two figures we find that there is value in including the AF model and we have retained them. We adjusted the main text to explain our rationale in the results section:

“When superposing a predicted model of full-length CTL1 on our experimentally determined CTL1 model, the prediction helps to approximate where unmodelled regions of the extracellular domain and M3/M4 protrude from the micelle, matching the map when displayed at lower contour level (Figure 4C)”.

and also in the discussion:

“By superposing an AlphaFold predicted model of the FLVCR2-Protein A chimera onto our model, it can be visualized that the protruding fiducial worked as intended (Figure 3C)”.

Figure 5: For PNS1, we agree that the AF model did not contribute meaningful information, and we have removed it.

Minor comments:

1. Figure 2C combines FLVCR1 and FLVCR2 despite large differences in signal magnitude, compressing data and obscuring trends. It would help the reader to interpret these results if the data were separated into independent panels. Overall, several of the figures are small and overly compressed, making interpretation difficult; resizing is recommended.

We agree. We have separated the data of FLVCR1 and FLVCR2 to avoid obscuring trends. We have also resized and edited Figure 2 for clarity. We find that the new version of Figure 2 is much improved.

2. The manuscript highlights putative sterol or lipid densities but provides no biochemical data to support their identity. Given the low contouring and uncertainty, these assignments should be toned down or removed.

We have toned down these assignments as suggested.

We are now writing for FLVCR2:

Line 135: *“The map shows elongated densities surrounding FLVCR2, possibly cholesterol-hemisuccinate (CHS) molecules or retained lipids, which were left unmodeled (Figure 3D).”*

And labels in Figure 3D have been adjusted to “putative sterols or lipids”.

We now write for CTL1:

Line 160: *“Multiple elongated densities, possibly sterols or retained lipids, can be seen surrounding M1, M5 and M6, at the interface with M7, and in a hydrophobic pocket at the back of the protein, formed by M1, M5 and M6 (Figure 4D).”*

and

Line 168: *“The outer surface of CTL1 is mostly hydrophobic, with a groove at the back where we observe the elongated densities in the map, while the putative binding pocket largely consists of hydrophilic residues (Figure 4E)”*

And labels in Fig 4D have been adjusted to “putative sterols or lipids”.

We now write for PNS1:

Line 184: *“Like CTL1, there seems to be elongated densities that could match lipids or sterols at the back of PNS1, opposite the entrance of the putative binding pocket (Figure 5D).”*

For the comparison of CTL1 and PNS1:

Line 202: *“PNS1 and CTL1 share 15% sequence identity, with only Tyr495_{CTL1} and Tyr394_{PNS1} conserved in the putative binding pocket (Figure 6C) and, intriguingly, most of the conserved residues form the hydrophobic pocket at the back, where the putative sterol/lipid densities are observed (Figure 6D and Figure S6).”*

And labels in Fig 6D have been adjusted to “putative sterol or lipid-binding groove”.

This concluding sentence has been deleted:

“The maps of CTL1 and PNS1 consistently show interactions with lipids and sterols within a cavity of conserved residues, and this could be important for their function.”

3. A workflow for the image processing of CTL1 should be included.

We have added this as the new Supplementary Figure 3.

4. Several of the figures employ highly complex color schemes, which makes it difficult to interpret the superpositions. For example, the features in Figure 6B-C are hard to distinguish.

We have simplified several panels (inc. 6B-C). We have also changed the representation of PNS1 and CTL1 in Figure 6B to more readily visualize the displacement of helices between the two structures. We find the new versions are easier to interpret.

Reviewer #2

In this manuscript, Nel et al. perform functional characterization of FLVCR1/FLVCR2/CTL1/PNS1/Cher1. They also determine the structures of FLVCR2, CTL1 and PNS1. Although generally scientifically sound, the manuscript does not present substantial new insights beyond what is already established in the literature. In instances where its findings diverge from published work, the experiments do not appear to have been conducted under conditions appropriate for a valid comparison.

We thank the reviewer for the review of our manuscript. We have addressed all the points raised as specified below. We find the new version of the manuscript is much improved.

Major points

- The manuscript does not examine choline transport across a range of pH conditions for CTL1. Given that CTL1 is a proton-coupled antiporter, omission of pH-dependent transport assays represents a substantial gap. As currently presented, the functional data cannot be considered conclusive and would explain why this observation differs from the majority of papers in the literature.

We have done pH-dependent uptake for CTL1, PNS1 and Cher1, but in the first version of the manuscript, this data was not included as no activity was present at any pH. We agree that this data is relevant for the discussion and should be included. We have now included this data in the paper in Figure S1 as panel E and F.

In short, we tested activity using two different concentrations of substrate, and at three different pH (5.5, 7.5 and 8.5), which overlaps with previous papers. In addition, we tested 100 μM substrate at pH 7.4 (fig 2A). In our hands, none of the transporters were active at the tested pH values:

We have included these observations in the results and discussion sections as follows:

Line 116: “In a range from pH 5.5 to pH 8.5, CTL1, PNS1 and Cher1 did not transport choline (Figure S1E and Figure S1F).”

Line 300: “In the current study, pH-dependency assays of CTL1, PNS1 and Cher1 were conducted with 0.5 μM and 2 μM choline (Figure S1E and Figure S1F), however none of these proteins transported choline above the level of the negative control. Since literature suggests some pH dependency for CTL1 (increased transport at pH 8.5) (27) and Cher1 (increased transport at pH 5.5) (24), it was not reported that these proteins were not inactive at physiological pH, which is why choline and ethanolamine uptake assays were conducted at pH 7.4 in the current study.”

We also provide the following perspective regarding previously published activity data on CTL1 and Cher1:

Inazu et al., 2013 (ref. 27, Figure 3A) used lung cancer cells to show uptake of choline using CTL1. Here the activity was pH dependent, with higher pH improving transport rates within the entire range tested (max pH 8.5). These assays thus display a surprising trend where high pH stimulates choline import.

Taylor et al., 2021 (ref. 31, Figure 2F), tested ethanolamine uptake (not choline) of CTL1 and CTL2 in human fibroblast cells. These assays also displayed this surprising trend where higher pH improving transport rates within the entire range tested (max pH 8.5). This led to an antiporter model suggested in the paper that CTL1 and CTL2 could be ethanolamine/proton antiporters (i.e. moving a proton away from the cytosol, unlike all other known antiporters).

Dettmer et al, 2014 (ref 24, Figure S3) reported choline transport in oocytes for Cher1 at pH 5 (1.75-fold higher than the water control) with retained activity at pH 7.4 (1.5 fold) and pH 7.7 (1.25 fold). Of note, Western blot or GFP-fluorescence was not used to check expression levels. Notably, here low pH (pH 5) improved uptake rates.

O'Regan and Meunier, 2003 (ref 29, Figure 3) did oocyte assays of *Torpedo marmorata* TmCTL. The pH used in the assays are not specified in the paper, but might be assumed to be pH 7.4-7.8 (standard pH in Barths solution). No test or quantification of the expression of TmCTL was provided. They observed choline uptake at 10 μ M choline that could be inhibited by the compound 3-HC.

- The FLVCR2 structure does not provide additional insight and is lower in resolution than previously published FLVCR2 structures. The rationale for including this dataset, and its incremental value to the field, requires clarification.

We include the structural data of FLVCR2 as a validation of the structures published in 2024-2025. The use of Protein A repeats as a fiducial marker is also of interest to the field.

To explain this, we have now modified the main text to read:

Line 275: *“When superposing our structure to other structures of FLVCR2 in the inward conformation, the RMSDCa is less than 1 Å (18,34). These structures are obtained either without the use of a fiducial, or by using an antibody or in our case, Protein A repeats. Even though our structure does not provide new information on FLVCR2, it confirms previous findings regarding choline binding and it supports that the use of Protein A did not perturb FLVCR2 in adopting the inward conformation, making it suitable for solving structures of small membrane proteins with pseudo-symmetry and no discernible features protruding from the micelle.”*

- No biochemical characterization/ purification figure of CTL1 is provided.

We have added the SDS-PAGE and SEC trace of CTL1s purification as part of the new Supplementary Figure 3:

- The CTL1 structure, despite its marginally improved resolution, does not appear to yield new mechanistic insight compared with the previously published model. While they identify the presence of a density that appears

as though it may not be choline, this density has not been characterized and therefore this does not provide any novel information.

The CTL1 model we have published is very similar to the previously published model. However, some regions, that could not be modelled in the previously published model, can now be included due to improved resolution (see Figure S4B). In addition, the CTL1 structure is unusual, given its suggested role in choline uptake, and thus there is value in the supporting result, presented here that the fold is correct. This is further supported by our inclusion of PNS1, which confirms the fold within the protein family.

- The FLVCR2 structure does not appear to provide additional mechanistic insight and is lower in resolution than previously published FLVCR2 structures. The rationale for including this dataset, and its incremental value to the field, requires clarification.

This point seems to be the same statement as the second major point made by the reviewer. Please see our reply above on how this point has been addressed.

- The oligomeric state of CTL1 remains unclear. Prior analyses have suggested it exists as a dimer. The authors should comment on whether their construct forms a monomer or higher-order assembly in solution or in the membrane mimic.

We comment on the oligomeric state we observed in the newly included Figure S3B and S3C and in the main text discussion as follows:

“One difference of note is that the previously determined CTL1 structure was a dimer, while we clearly observe a monomeric state in both SDS-PAGE gels, SEC profile and micrographs. It remains unclear whether a dimeric or monomeric form is the physiological state of CTL1 and experimental protocol differences might explain the discrepancy.

We determined the structure of yeast PNS1 by crystallography with experimental phasing to verify the unusual fold of human CTL1 and confirm that is representative of CTL family members. Notably, PNS1 does not contain the extracellular domain found in CTL1 but it does adopt the same overall conformation and is also in a monomeric state.”

and further discussed in the conclusion:

“Based on structural similarity to PNS1, CysZ, a prokaryotic membrane permease of sulphate (SO_4^{2-}) utilized in cystine biosynthesis, was an unexpected result Figure S7A). CysZ is proposed to adopt a hexameric arrangement as a trimer of dimers with each monomer having 6 helices that span the membrane to differing extents (41). Their proposed membrane topology thus differs significantly from the topology and oligomerization state we observe for PNS1 and CTL1, and it is hard to reconcile with any functional similarity.”

In more detail: The expected size of monomeric CTL1 is ~73 kDa (Uniprot: Q8WW15). Membrane proteins migrate faster in SDS-PAGE gels due to their hydrophobic properties. The SDS-PAGE gel shows a distinct band at ~60 kDa, also confirmed by MS to be CTL1. We do not observe a band that would correspond to a dimer, but any oligomeric state could be disassociated due to the SDS. NB. note that membrane protein samples are not boiled before SDS-PAGE loading as this would lead to aggregation.

The SEC profile shows three peaks; at the void volume (8 mL), a weak peak at 10 mL that could correspond to a dimer and a main peak at 11 mL corresponding to the monomeric form in a detergent micelle.

2D classes obtained during cryo-EM data processing were consistently of a single CTL1 molecule within the micelle.

Since the first CTL1 structure solved in 2022 (ref 17: Xie T, et al. doi:10.1016/j.str.2022.05.015) was in a final SEC buffer that contained a mixture of DDM, LMNG, GDN detergents, CHS and no HC-3, the dimerization they observe might have been the result of this unique, large micellar environment. In the end our experimental protocols differ for previously published work, and this likely explain the discrepancy.

- Method details for peptidisc preparation/incorporation are missing. Without these, reproducibility and interpretation of the structural data are limited.

We thank the reviewer for catching this omission. We have added a paragraph detailing this in the Methods section “*Purification of CTL1 for cryo-EM*”.

- The title seems unusual seeing as though the authors are claiming that 2/3 of the transporters studied are not actually choline transporters.

We appreciate this point. We have adjusted the title which now reads:
“*Structure-function analysis of proposed choline transporter families FLVCR and CTL*”

Reviewer #3

The manuscript entitled "Biochemical and structural comparison of eukaryotic choline transport protein families FLVCR and CTL" by Nel et al is a careful examination of choline transport (or lack thereof) by different transmembrane proteins. A major conclusion of this paper is that contrary to previous reports, CTL1-type proteins do not function as choline transporters, at least in the experimental conditions tested here. Impressively, the authors also report structures of human CTL1, FLVCR2 (cryo-EM) and the yeast CTL1 homologue, PNS1 (crystallography). The structure of human CTL1 was reported previously, in a dimeric configuration, whereas here the authors show a monomer at an improved resolution. The structure of FLVCR2 in this conformation has also been reported before, though this structure here demonstrates the use of Protein A as an inbuilt fiducial, and fits well with the logical progression of the paper. The unusual fold of human CTL1 observed is verified by the crystal structure determination of a homologous protein, PNS1, from yeast. This supports that the quite fascinating open-to-the-membrane, bowl-like structure is conserved in this family. Although the authors are yet to uncover the mechanistic details of CTL1 function, I think this work a valuable contribution to the field, and the paper to be well-written & presented.

We thank the reviewer for the review of our manuscript. We have addressed all points raised and find the new version is much improved.

I do have several comments about the structural analysis of CTL1, that I think should be addressed before publication:

Regarding CTL1 in MNG vs peptidisc, I am not sure that there are no "observed differences between the two maps" (Line 151)-when I aligned the maps EMDB-50251 and EMDB-50252 it seems they align well on parts of the N-terminal portion of the transmembrane protein, but the M3/M4 helices in the peptidisc structure seem to have been pulled inward closer to the N-terminal portion of the protein. Could this possibly be a peptidisc mediated artefact? Could the authors clarify this, as although these differences are only visible at lower contours, they are still apparent in the maps. An overlay figure could be good here.

We thank the reviewer for this analysis. As suggested, we now include an overlay figure as a panel in Figure S3:

We have also modified the main text to elaborate on the difference between the two maps:

Line 151 *"The core of the volumes are the same with the only noticeable difference in the positioning of M3/M4, which is compressed in the map with peptidisc (map-map cross-correlation coefficient = ~0.9; Figure S3G), thus we proceeded with the analysis using the sample in LMNG."*

In the deposited model the authors chose not to model the extracellular domain, or the M3 and M4 helices. While I understand that the resolution is lower here, from inspection of the maps, it seems that placement of the extracellular domain is feasible. This could be achieved with the AlphaFold model and Isolde. I think that it would increase the utility of the structure. The authors might consider including this domain (and perhaps m3/m4) in the PDB deposition, perhaps with appropriately conservative modelling (truncate sidechains). DeepEMhancer or the EMReady2 web server might be used to aid modelling.

We thank the reviewer for these suggestions and careful examination of the maps. The model of CTL1 was initiated from an AlphaFold model prediction, which was fitted into the map with Isolde. When moving to

Coot/phenix to build and refine the model, we ultimately deleted most of the extracellular domain and M3/M4 since the main chain was difficult to trace and we did not trust the output. In particular, M3/M4 could not be modelled with any confidence since only a small region in the middle of the helices was well-defined. As for the extracellular domain, refinements distorted the main chain by placing it in map regions of the surrounding micelle. In the end we concluded that the most honest thing would be to delete these regions to emphasize that no experimental evidence is available to support the AF prediction.

Did the authors try focused refinements with a mask around the N-terminal & extracellular domain, and/or 3DVA? Further to this-there is no workflow for CTL1. Even if similar to FLVCR2, the authors really should include a separate workflow, with representative classes, # of particles, and images of maps. I also think an SDS-PAGE gel should be shown with the workflow, and a SEC trace. Was there any hints of a dimer? It seems that PNS1 might have a monomer-dimer self-association per the gel and SEC trace in the Supplementary. I note again the previous structure of CTL1 was a dimer-I wonder if this is functionally relevant.

We appreciate the reviewer's comment regarding the processing of the data and we can elaborate on the suggested jobs with Figure S3F. We used 3D classification to tease out particles where the extracellular domain and M3/M4 were most prominent. This class was used for NU-refinement and subsequent local refinements on the extracellular domain or M3/M4. However, from the local resolution estimation, the quality of the map still quickly worsened to ~5 Å.

We also thank the reviewer for bringing this up and we can comment on the oligomeric state we observed with the newly included Figure S3B and S3C. The expected size of monomeric CTL1 is ~73 kDa (Uniprot: Q8WWI5) and the SDS-PAGE gel shows a distinct band at ~58 kDa and a faint band at ~73 kDa. It is known that membrane proteins can migrate faster in SDS-PAGE gels and have a seemingly smaller size. However, we did not observe a band that would correspond to a dimer between 166-150 kDa. The protein at ~73 kDa could also be a contaminant, which was separated after SEC.

The SEC profile shows three peaks; at 8 mL is where we usually expect aggregates (void volume), at 10 mL oligomeric states that could correspond to a dimer and at 11 mL the monomeric form, which was the most abundant.

Additionally, 2D classes obtained during cryo-EM data processing were consistently of a single CTL1 molecule within the micelle.

Since the first CTL1 structure solved in 2022 was in a final SEC buffer that contained a mixture of DDM, LMNG, GDN detergents, CHS and no HC-3, the dimerization could have been the result of this unique, large micellar environment.

The functional characterisation in oocytes is convincing, with the uptake assay data in Figure 2 clear and presented very nicely (albeit a little small and hard to read, this should be addressed in the final version prior to publication). These data are strong evidence against CTL1 being a choline transporter, although, I wondered why pH/Na⁺ dependence of choline/ethanolamine uptake was not assayed for CTL1? It does seem unlikely that these are coupled symporters/antiporters given the architecture, however, given claims by Taylor et al, JBC (2021), that CTL1 functions as a Na⁺-independent, H⁺/ethanolamine antiporter, it seems that this should be tested, or at least be caveated in the discussion.

We agree that the previous version of Figure 2 was not optimal and we have rearranged the panels to facilitate interpretation.

A sodium-dependency assay was not done for CTL1, PNS1 or Cher1 since transport of choline was not observed in initial experiments that contain 96 mM NaCl. We added a sentence in the discussion to explain why we did not conduct this assay:

“Since CTL1, PNS1 and Cher1 did not transport choline or ethanolamine in standard ND96 buffer with 96 mM NaCl, sodium-dependency assays were not conducted.”

We have done pH-dependent uptake for CTL1, PNS1 and Cher1, but in the first version of the manuscript, this data was not included as no activity was present at any pH. We agree that this data is relevant for the discussion and should be included. We have now included this data in the paper in Figure S1 as panel E and F.

In short, we tested activity using two different concentrations of substrate (0.5 μM and 2 μM choline). We tested 3 different pH (5.5, 7.5 and 8.5), which overlaps with previous papers. In our hands, none of the transporters are active at the tested pH values:

We have included these observations in the results and discussion sections as follows:

Line 116: “In a range from pH 5.5 to pH 8.5, CTL1, PNS1 and Cher1 did not transport choline (Figure S1E and Figure S1F).”

Line 300: “In the current study, pH-dependency assays of CTL1, PNS1 and Cher1 were conducted with 0.5 μM and 2 μM choline (Figure S1E and Figure S1F), however none of these proteins transported choline above the level of the negative control. Since literature suggests some pH dependency for CTL1 (increased transport at pH 8.5) (27) and Cher1 (increased transport at pH 5.5) (24), it was not reported that these proteins were not inactive at physiological pH, which is why choline and ethanolamine uptake assays were conducted at pH 7.4 in the current study.”

We also provide the following perspective regarding previously published activity data on CTL1 and Cher1:

Inazu et al., 2013 (ref. 27, Figure 3A) used lung cancer cells to show uptake of choline using CTL1. Here the activity was pH dependent, with higher pH improving transport rates within the entire range tested (max pH 8.5). These assays thus display a surprising trend where high pH stimulate choline import.

Taylor et al., 2021 (ref. 31, Figure 2F), tested ethanolamine uptake (not choline) of CTL1 and CTL2 in human fibroblast cells. These assays also displayed this surprising trend where higher pH improving transport rates within the entire range tested (max pH 8.5). This led to an antiporter model suggested in the paper that CTL1 and CTL2 could be ethanolamine/proton antiporters (i.e. moving a proton away from the cytosol, unlike all other known antiporters).

Dettmer et al, 2014 (ref 24, Figure S3) reported choline transport in oocytes for Cher1 at pH 5 (1.75-fold higher than the water control) with retained activity at pH 7.4 (1.5 fold) and pH 7.7 (1.25 fold). Of note, Western blot or GFP-fluorescence was not used to check expression levels. Notably, here low pH (pH 5) improved uptake rates.

O’Regan and Meunier, 2003 (ref 29, Figure 3) did oocyte assays of *Torpedo marmorata* TmCTL. The pH used in the assays are not specified in the paper, but might be assumed to be pH 7.4-7.8 (standard pH in Barths solution). No test or quantification of the expression of TmCTL was provided. They observed choline uptake at 10 μM choline that could be inhibited by the compound 3-HC.

"Our data suggests that CTL proteins like PNS1, CTL1 and CherI do not bind choline and are not choline transporters." While I agree that the authors have shown these proteins are unlikely to be choline transporters, is it not possible that choline or choline containing molecules may bind but not be transported? This could be toned down. Further to this, if the authors are suggesting CTL proteins are not transporters, perhaps they should consider removing "choline transport" from the title.

We agree and we have toned down the sentence which now reads:

Line 346: *"Our data suggests that CTL proteins like PNS1, CTL1 and CherI do not transport choline but it cannot be discounted that these proteins might bind choline to function."*

Supporting this, we have also adjusted the title which now reads:

"Structure-function analysis of proposed choline transporter families FLVCR and CTL"

Line 541: "mano-olin"-should this be mono-olein?

We have corrected this mistake.

March 31, 2026

RE: Life Science Alliance Manuscript #LSA-2025-03583R

Dr. Bjørn Panyella Pedersen
Aarhus University
Department of Molecular Biology and Genetics
Universitetsbyen 81
MBG-AU
Aarhus C, Danmark 8000
Denmark

Dear Dr. Pedersen,

Thank you for submitting your revised manuscript entitled "Structure-function analysis of proposed choline transporter families FLVCR and CTL". Your revised manuscript was assessed by all the original reviewers whose comments are appended below. Although all the reviewers agreed that the revised version addressed many of their previous concerns, they have some outstanding concerns.

Reviewers 1 and 2 expressed concerns on the concentration of choline used, in comparison with published Km values for CTL1. Moreover Reviewer 2 also highlights that the cross-experimental comparisons are not done under consistent conditions. We agree with Reviewer 2 that you must clearly address these two issues with changes to the text. Another significant concern raised by Reviewer 2 is on the integrity of purified CTL1 protein, which we agree must be demonstrated.

Overall, we would be happy to publish your paper in Life Science Alliance pending resolution of the above points and final revisions necessary to meet our formatting guidelines.

We request you to submit a revised manuscript document with all changes highlighted.

MANUSCRIPT ORGANIZATION AND FORMATTING:

To avoid unnecessary delays in the acceptance and publication of your paper, please read the following information carefully. Full guidelines are available on our Instructions for Authors page, <https://www.life-science-alliance.org/authors>

- Please address the minor comments of Reviewer 3 related to the title of this study and minor typos.
- Kindly remove the eTOC summary and suggested post for the social media from the title page.
- We request you to include sub-headings in the results section.
- You have provided reference number 48 in conjunction with the yeast-expression optimised sequence for FLVCR2, PNS1 and CTL1. Please ensure that these details are present in the cited reference, otherwise please provide details for yeast-expression optimised sequences.
- Please provide full details for peptidisc generation or include a source for peptide(s).
- For your studies that report experiments with vertebrate material, please follow our guidelines (<https://www.life-science-alliance.org/editorial-policies#animals>) to confirm that all experiments were performed in accordance with relevant guidelines and regulations, and identify the institutional and/or licensing committee approving the experiments.
- Kindly provide sufficient details of the size exclusion chromatography and SDS-PAGE methods, the outcome of which are presented in S2-S4.
Please add the X and Bluesky handles of your host institute/organization, as well as your own, and/or one of the authors, in our system
- Please consult our manuscript preparation guidelines <https://www.life-science-alliance.org/manuscript-prep> and make sure your manuscript sections are in the correct order
- The contributions selected for Ronja Driller do not qualify them for authorship. Please either update the contributions in our system and in the Author Contributions section of the manuscript, or let us know if the author needs to be removed (and added eventually to the acknowledgment section).
- Please add callouts for Figures S2A-F and S4A-D to your main manuscript text.
- Please be sure that the authorship listing and order is correct

per figure for this information. These files will be linked as supplementary "Source Data" files.

We welcome submissions of potential cover images for the issue of LSA in which your work would appear. If you have high quality images associated with this work, please feel free to email these, with a caption, to the journal office.

LSA encourages authors to provide a 30-60 second video where the study is briefly explained. We will use these videos on social media to promote the published paper and the presenting author (for examples, see <https://docs.google.com/document/d/1-UWCfbE4pGcDdcgzcmiuJI2XMBJnxKYeqRvLLrLSo8s/edit?usp=sharing>). Corresponding or first-authors are welcome to submit the video. Please submit only one video per manuscript. The video can be emailed to contact@life-science-alliance.org

FINAL FILES:

The following items are required for acceptance.

The license to publish form must be signed before your manuscript can be sent to production. A link to the license to publish form will be available to the corresponding author only. Please take a moment to check your funder requirements.

Thank you for your attention to these final processing requirements. Please revise and format the manuscript and upload materials as soon as you are able.

Thank you for this interesting contribution to the literature. We look forward to publishing your paper in Life Science Alliance.

Sincerely,

Sarita Hebbar, PhD
Scientific Editor
Life Science Alliance
<http://www.lsajournal.org>

Reviewer #1 (Comments to the Authors (Required)):

The authors have addressed many of my concerns, but the presentation of the transport assay could be improved. A more complete characterization of choline transport by CTL1 would be very informative. This characterization should include concentrations above the reported Km for CTL1, which would help to clarify its physiological role.

Reviewer #2 (Comments to the Authors (Required)):

While the authors have addressed many of the previous reviewer comments, several important concerns remain outstanding, and there are notable inadequacies in how key experiments have been performed that must be addressed before the conclusions of this study can be accepted.

The proton dependence assay for CTL1 was conducted at choline concentrations of only 0.5 and 2 μM . Given the previously proposed $K_{0.5}$ for this transporter, these concentrations appear insufficient to draw reliable conclusions. The authors should repeat this experiment at 100 μM choline, consistent with the approach taken at pH 7.5. The absence of choline uptake observed at pH 7.5 with 100 μM choline may be explained by the lack of a proton gradient under those conditions, rather than by any intrinsic property of the transporter - an important distinction that the current experimental design does not adequately address. This experimental set-up for CTL1 is inconsistent with the FLVCR1/2 experiments, in which 50 μM choline was used. The authors should ensure that choline concentrations are applied consistently across all transport assays to permit meaningful cross-experimental comparisons. Additionally, uptake assays for CTL1 should be performed in the presence and absence of the protonophore CCCP, again at an appropriately high choline concentration (e.g., 100 μM), the same as was performed for FLVCR1/2. Without conducting these experiments in a consistent, thorough, and rigorously controlled manner, the conclusion that CTL1 does not transport choline remains unsubstantiated - particularly in light of the existing literature supporting a transport function for this protein. This concern is further compounded by the observation that the authors do not detect efficient choline transport by FLVCR1, despite this protein now being established as a bona fide choline transporter with molecular determinants clearly defined by cryo-EM. This raises broader questions about the sensitivity and reliability of the choline transport assay system employed in this study, which the authors should address directly.

Comment 2: Protein integrity

The SDS-PAGE and SEC profiles raise concerns regarding the integrity of the purified CTL1 protein. There are multiple peaks observed on the SEC, and bands on the gel. While it is well established that membrane proteins can exhibit anomalously fast migration on SDS-PAGE, the observed band at approximately 55-60 kDa is substantially lower than the predicted molecular weight of 73 kDa, and differs considerably from the CTL1 band position reported in the western blot data of Iwao et al. Of particular concern is the presence of a distinct band just above the 66 kDa marker, which may represent intact CTL1, suggesting that the predominant lower band at ~60 kDa corresponds to a degradation product. If this is the case, the majority of the purified material may not be full-length protein, which would have significant implications for the functional and structural conclusions drawn in this study. Could the authors please confirm integrity of these two bands by mass spectrometry analysis to clarify whether their CTL1 is predominantly intact or not.

Reviewer #3 (Comments to the Authors (Required)):

The revisions by the authors have improved the manuscript and it is now publishable in my opinion. I am satisfied with the clarifications regarding structural biology, and agree with the inclusion of the pH dependent uptake assays for CTL1, PNS1 and Cher1- this adds weight to the suggestion that CTL1 does not function as previously thought.

The title could still be improved. The FLVCR family are bona fide choline transporters, so probably shouldn't be called "proposed".

Some small errors remain that should be fixed, e.g. methionine is spelled incorrectly in Figure 1.

Abstract- line 26 should read "appears"

Reviewers' Comments

(authors' reply in blue)

Reviewer #1

The authors have addressed many of my concerns, but the presentation of the transport assay could be improved. A more complete characterization of choline transport by CTL1 would be very informative. This characterization should include concentrations above the reported K_m for CTL1, which would help to clarify its physiological role.

We appreciate the continued input toward improving the manuscript.

We tested all five proteins (FLVCR1, FLVCR2, CTL1, PNS1 and CherI) with 100 μ M choline and ethanolamine at pH 7.4 (Figure 2A and 2B).

This assay was at 10x above the K_m value for CTL1 (~ 10 μ M) reported in the literature. For this uptake assay, repeated several times, CTL1, along with PNS1 and CherI, did not exhibit any activity.

We agree that it would make sense to use higher concentrations of choline for the pH-dependency assay for CTL1 (Figure S1E and S1F). However, since no activity was observed at 100 μ M at physiological pH, we did not pursue this experiment further, due to workload and expense considerations (radiolabeled choline).

Literature suggests that if CTL1 transports choline, it would do so in a pH range of 6.0-8.5 (Inazu et al., 2013) while CherI (with a proposed K_m in the 7-9 nM range(!)) was reported to have activity at pH 5.0-7.7 (Dettmer et al., 2014). I.e. if there was activity we should be able to see it at pH 7.4 for both CTL1 and CherI.

We have toned down the discussion of the activity results re CTL1 further and explained why additional assays regarding pH-dependency at higher choline concentrations were not conducted.

In the discussion:

“In the current study, pH-dependency assays of CTL1, PNS1 and CherI were conducted with 0.5 μ M and 2 μ M choline, while initial activity tests were done at 100 μ M (Figure S1E and Figure S1F) None of these proteins displayed activity above the level of the negative control for any of these conditions in our hands, even using 100 μ M choline.

PNS1 have no previous reports of transport activity, while CherI has been reported to have a K_m in the low nM range, and would have been expected to display activity. For CTL1, the pH assays are done at concentrations matching the first reported K_m (1 μ M for iCTL1) but below the later reported K_m for human CTL1 of 10 μ M (22,27). However as no transport was observed using 100 μ M choline at pH 7.4, in contrast to what was reported by Inazu et al. (27) (Figure 2A), we did not conduct additional pH-dependency assays for any of the CTL proteins.

As CTL1, PNS1 and CherI did not show any activity for either choline or ethanolamine in standard ND96 buffer with 96 mM NaCl, further sodium-dependency assays were also not done.

Under our assay conditions, only FLVCR1 and FLVCR2 showed significant transport of choline and ethanolamine (Figure 2A and Figure 2B), leading to the speculation that CTL proteins might not directly transport these compounds. This has also been reported and speculated by others (25,33). In particular our results coincide with those of Nguyen and co-workers, who could not observe choline import by CTL1 or CTL2 in HEK cells (33).”

In the conclusion:

“Our data show that CTL proteins CTL1, PNS1 and CherI do not transport choline in oocytes under the conditions tested, but it cannot be discounted that these proteins might have some transport functionality under different experimental conditions. The weak structural similarity of CTL proteins to SLC25 could suggest a role as exchangers, but their closed bowl-like shape does not seem to support any known transport mechanism. Thus, we speculate that CTL proteins could possibly be involved in unidentified regulatory functions related to choline metabolism.”

Reviewer #2

While the authors have addressed many of the previous reviewer comments, several important concerns remain outstanding, and there are notable inadequacies in how key experiments have been performed that must be addressed before the conclusions of this study can be accepted.

We thank the reviewer for their continued input towards improving the manuscript.

The proton dependence assay for CTL1 was conducted at choline concentrations of only 0.5 and 2 μM . Given the previously proposed $K_{0.5}$ for this transporter, these concentrations appear insufficient to draw reliable conclusions. The authors should repeat this experiment at 100 μM choline, consistent with the approach taken at pH 7.5. The absence of choline uptake observed at pH 7.5 with 100 μM choline may be explained by the lack of a proton gradient under those conditions, rather than by any intrinsic property of the transporter - an important distinction that the current experimental design does not adequately address.

We tested all five proteins (FLVCR1, FLVCR2, CTL1, PNS1 and CherI) with 100 μM choline and ethanolamine at pH 7.4 (Figure 2A and 2B).

This assay was at 10x above the K_m value for CTL1 (~10 μM) reported in the literature. For this uptake assay, repeated several times, CTL1, along with PNS1 and CherI, did not exhibit any activity.

We agree that it would make sense to use higher concentrations of choline for the pH-dependency assay for CTL1 (Figure S1E and S1F). However, since no activity was observed at 100 μM at physiological pH, we did not pursue this experiment further, due to workload and expense considerations (radiolabeled choline).

Literature suggests that if CTL1 transports choline, it would do so in a pH range of 6.0-8.5 (Inazu et al., 2013) while CherI (with a proposed K_m in the nM range) was reported to have some activity at pH 5.0-7.7 (Dettmer et al., 2014). I.e. if there was activity we should be able to see it at pH 7.4 for both CTL1 and CherI.

We have toned down the discussion of the activity results re CTL1 further and explained why additional assays regarding pH-dependency at higher choline concentrations were not conducted.

In the discussion:

“In the current study, pH-dependency assays of CTL1, PNS1 and CherI were conducted with 0.5 μM and 2 μM choline, while initial activity tests were done at 100 μM (Figure S1E and Figure S1F) None of these proteins displayed activity above the level of the negative control for any of these conditions in our hands, even using 100 μM choline.

PNS1 have no previous reports of transport activity, while CherI has been reported to have a K_m in the low nM range, and would have been expected to display activity. For CTL1, the pH assays are done at concentrations matching the first reported K_m (1 μM for tCTL1) but below the later reported K_m for human CTL1 of 10 μM (22,27). However as no transport was observed using 100 μM choline at pH 7.4, in contrast to what was reported by Inazu et al. (27) (Figure 2A), we did not conduct additional pH-dependency assays for any of the CTL proteins.

As CTL1, PNS1 and CherI did not show any activity for either choline or ethanolamine in standard ND96 buffer with 96 mM NaCl, further sodium-dependency assays were also not done.

Under our assay conditions, only FLVCR1 and FLVCR2 showed significant transport of choline and ethanolamine (Figure 2A and Figure 2B), leading to the speculation that CTL proteins might not directly transport these compounds. This has also been reported and speculated by others (25,33). In particular our results coincide with those of Nguyen and co-workers, who could not observe choline import by CTL1 or CTL2 in HEK cells (33).”

In the conclusion:

“Our data show that CTL proteins CTL1, PNS1 and CherI do not transport choline in oocytes under the conditions tested, but it cannot be discounted that these proteins might have some transport functionality under different experimental conditions. The weak structural similarity of CTL proteins to SLC25 could suggest a role as exchangers, but their closed bowl-like shape does not seem to support any known transport mechanism. Thus, we speculate that CTL proteins could possibly be involved in unidentified regulatory functions related to choline metabolism.”

This experimental set-up for CTL1 is inconsistent with the FLVCR1/2 experiments, in which 50 μM choline was used. The authors should ensure that choline concentrations are applied consistently across all transport assays to permit meaningful cross-experimental comparisons.

We appreciate this point. Our reasoning is as follows: We have ensured that within each protein family (FLVCR or CTL) we have applied all experimental parameters consistently. In some cases, there are variations as noted. This is because we continued the characterization of the FLVCR family after we observed strong and reproducible activity of these proteins. We deemed it unnecessary to include the CTL proteins in these experiments since we could not establish any activity. Throughout the text and in the figures we strive to be very clear about the concentrations used to facilitate readability and proper cross-experimental comparisons.

Additionally, uptake assays for CTL1 should be performed in the presence and absence of the protonophore CCCP, again at an appropriately high choline concentration (e.g., 100 μM), the same as was performed for FLVCR1/2.

We disagree with this. The experiment with FLVCR1 and FLVCR2 using CCCP (Figure 2G) was done because clear activity was observed using 100 μM choline (Figure 2A). Since no activity was observed for CTL1, PNS1 and CherI with 100 μM choline, there is no expectation that this lack of activity would be altered by the addition of CCCP, which should inhibit activity, not augment it.

Without conducting these experiments in a consistent, thorough, and rigorously controlled manner, the conclusion that CTL1 does not transport choline remains unsubstantiated - particularly in light of the existing literature supporting a transport function for this protein.

We have toned down the results from the activity assays further in the discussion.

We reiterate that some literature support our findings. They remain in agreement with Zufferey et al. 2004 (<https://doi.org/10.1023/b:nere.0000013752.43906.e5>), that examine tCTL1 and could not use it to restore choline uptake. We cite:

“Choline transport assays in *hnm1-ise tCTL1* strain [with a knockout of the endogenous choline transporter] indicated that tCtl1p has a similar affinity for choline as the endogenous choline transporter Hnm1p (12). Our finding that yeast possesses a homolog of tCtl1p, Pns1p, prompted us to question the validity of the transport hypothesis. Our studies show that disruption of *PNS1* gene in wild-type or *hnm1 Δ* strains has no effect on choline uptake. Furthermore overexpression of *PNS1* or *tCTL1* in *hnm1 Δ* , lacking the only yeast choline transporter, did not restore choline uptake. Together, these data indicate that Pns1p, tCtl1p and probably all members of this family of membrane proteins are not choline transporters.”

And also with Nguyen et al. 2024 (<https://doi.org/10.1038/s41422-023-00923-y>) who could not observe choline import by CTL1(SLC44a1) or CTL2(SLC44a2) in HEK cells (33). We cite:

“SLC44a1 and SLC44a2 have been reported to be choline transporters. However, we were unable to show choline import activity for these transporters under the tested conditions (Supplementary information, Fig. S4c, d).”

For convenience we paste their Fig S4c,d here:

This concern is further compounded by the observation that the authors do not detect efficient choline transport by FLVCR1, despite this protein now being established as a bona fide choline transporter with molecular determinants clearly defined by cryo-EM.

This is a misunderstanding of our findings, and we have revised the main text to better explain our results. In fact we observe strong activity from FLVCR1. The strength of the comparison within the same protein family in the same experimental setup is that we can better judge the difference between isoforms. In this case, indeed, FLVCR2 is more active than FLVCR1 when looking at uptake using 100 μM choline (Figure 2A). More specifically, FLVCR2 exhibits “low affinity, high V_{max} ” transport of choline, (Figure 2D) vs FLVCR1 that has comparably “higher affinity and lower V_{max} ” kinetics (Figure 2C). In the discussion, we summarize this by stating “*that FLVCR1 shows about six times stronger apparent affinity for choline compared to FLVCR2*”.

We also compare our results in the discussion with what has been found in literature:

“We determine an apparent affinity (K_m) of $\sim 26 \mu\text{M}$ for **FLVCR1**, which lies in-between the values observed by Ri and co-workers (approximately $47 \mu\text{M}$) and Son and co-workers ($\sim 8 \mu\text{M}$), who used HEK cell assays (18,35). The K_m of **FLVCR2** for choline is around $164 \mu\text{M}$, which is ~ 2.5 times weaker compared to that obtained by Ri and co-workers ($K_m \sim 64 \mu\text{M}$) in HEK cells (18). This stronger apparent affinity in HEK cells could be due to the FLVCR proteins being co-expressed with choline kinase A, leading to a decrease in the local concentration of choline by shunting the equilibrium towards phosphocholine (18). Another study by Nguyen and co-workers also co-expressed FLVCR2 with choline kinase A in HEK cells and obtained a K_m of around $100 \mu\text{M}$ (33), which is a value comparable to the result we obtained”.

There is no discrepancy between our observations and the published data, and we also state in the conclusion that we confirm FLVCR1 and FLVCR2 to be choline transporters.

This raises broader questions about the sensitivity and reliability of the choline transport assay system employed in this study, which the authors should address directly.

We have toned down the results from the activity assays further and explained why additional assays regarding pH-dependency at higher choline concentrations were not conducted:

In the discussion:

“In the current study, pH-dependency assays of CTL1, PNS1 and Cher1 were conducted with 0.5 μM and 2 μM choline (Figure S1E and Figure S1F), however as none of these proteins transported choline above the level of the negative control at 100 μM choline (Figure 2A), no further pH-dependency assays were done for CTL proteins?”

In the conclusion:

“Our data show that CTL proteins CTL1, PNS1 and Cher1 do not transport choline in oocytes under the conditions tested, but it cannot be discounted that these proteins might have some transport functionality under different experimental conditions. The weak structural similarity of CTL proteins to SLC25 could suggest a role as exchangers, but their closed bowl-like shape does not seem to support any known transport mechanism. Thus, we speculate that CTL proteins could possibly be involved in unidentified regulatory functions related to choline metabolism.”

Comment 2: Protein integrity

The SDS-PAGE and SEC profiles raise concerns regarding the integrity of the purified CTL1 protein. There are multiple peaks observed on the SEC, and bands on the gel. While it is well established that membrane proteins can exhibit anomalously fast migration on SDS-PAGE, the observed band at approximately 55-60 kDa is substantially lower than the predicted molecular weight of 73 kDa, and differs considerably from the CTL1 band position reported in the western blot data of Iwao et al.

The Western blots of CTL1 from tissue lysate obtained by Inazu et al., 2013 (Figure 5A) and Iwao et al., 2016 (Figure 4B) were derived from 10% SDS-PAGE gels, while the gel we present shows purified CTL1 on a gradient 4-20% Bis-Tris SDS-PAGE gel.

Notably, the purification that yielded the structure of CTL1 by Xie et al (Figure S4B in Xie et al., 2022; <https://doi.org/10.1016/j.str.2022.05.015>) shows an SDS-PAGE gel, of an unknown percentage, with multiple bands between 55-72 kDa. Due to the insect cell expression system (SF9) used in our study, it is possible that the glycosylation pattern differs from the mammalian (HEK) expression system used in other studies. Glycosylation prediction software predict N135, N180 and N507 as possible glycosylation sites. N507 is located on the cytosol side preventing glycosylation. N135 and N180 are part of the extracellular domain of CTL1, which is poorly resolved in all published maps, so it is not possible to determine if these residues are glycosylated based on cryo-EM maps.

```
Name: ct11      Length: 657
MGCCSSASSAAQSSKREWKPLEDRSCTDIPWLLLFILFCIGMGFICGFSIATGAAARLVSGYDSYGNICGQKNTKLEAIP
NSGMDHTQRKYVFFLDPCNLDLNRKIKSVALCVAACPRQELKTLSDVQKFAEINGSALCSYNLKPSEYTTSPKSSVLCP
KLPVPASAPIPFHRCAPVNSICYAKFAEALITFVSDNSVLHRLISGVMTSKEIILGLCLLSLVL SMLMVIIRYISRLV
VWILITLVLGSLGGTGVLWNLVYAKQRRSPKETVPEQLQTAEDNLRALLIYAISATVFTVILFLIMLVMRKRVALTIAL
FHVAGKVF IHLPLLVGFQPFWTFALVLFVWVYIMTLFLFGTTGSPVQNEQGFVEFKISGPLQYMIWYHVVGLIWISEFIL
ACQQMTVAGAVTYVYTRDKRNL PFTPI LASVNRLLIRYHLGTVAKGSFIITLVKIPRMLMYIHSQLKGKENACARCVLK
SCICCLWCLEKCLNYLNQNAVYATAINSTNFCTSAKDAFVILVENALRVATINTVGDFFMLFLGKVLIVCSTGLAGIMLLN
YQDDYTVWVLP LLIIVCLFAFLVAHCFLSIYEMVVDVLF LCF AIDTKYNDGSPGREFYMDKVLMEFVENS RKAMKEAGKGG
VADSRLEKPMASGASSA
```

```
.....N.....
.....N.....
.....N.....
```

(Threshold=0.5)

SeqName	Position	Potential	Jury	N-Glyc
		agreement		result
ct11	135 NGS A	0.5232	(7/9)	+
ct11	180 NIS C	0.6537	(8/9)	+
ct11	507 NST N	0.6000	(6/9)	+

Deglycosylation tests of our CTL1 sample show that the molecular weight does not change with the addition of deglycosylases, suggesting that in SF9 cells the sample is not glycosylated. Whether it is so in HEK cell expression systems has not been examined.

Of particular concern is the presence of a distinct band just above the 66 kDa marker, which may represent intact CTL1, suggesting that the predominant lower band at ~60 kDa corresponds to a degradation product. If this is the case, the majority of the purified material may not be full-length protein, which would have significant implications for the functional and structural conclusions drawn in this study.

For functional queries, quality of the purified protein sample does not apply since the construct used for oocytes contains the full-length ORF of CTL1, and is fused to GFP at the C-terminus. The oocytes translate the RNA into full-length CTL1 supported by confocal microscopy images that show fluorescence on the plasma membrane of the oocytes (Figure S1B).

Regarding the integrity of the sample for structural studies, it is correct that the lower bands on the gel could represent nicks in the protein chain and fragmented CTL1. However, the cryo-EM data collected shows that the particles do not represent fragments of CTL1 (Figure S3F) since there is density for the extracellular domain as well as M3/M4 that protrude from the micelle (Figure 4C). The extracellular domain and loop connecting M3/M4 are not built, because of lower local resolution, but they are present in the map at a low threshold.

We use the sample only to obtain cryo-EM maps and these maps are of sufficient quality to build the CTL1 model. This model superposes well with AF predictions and the model from the publication from Xie et al., 2022. The purification we present is of equal quality to their work with CTL1 (Figure S4B in Xie et al., 2022; <https://doi.org/10.1016/j.str.2022.05.015>). Cryo-EM Maps of our sample and the Xie et al sample both show the extracellular domain and the models remain in the same conformation, with an RMSD of 0.8 Å (Figure S5B).

Could the authors please confirm integrity of these two bands by mass spectrometry analysis to clarify whether their CTL1 is predominantly intact or not.

While mass spectrometry (MS) is used routinely by us to confirm the identity of the target protein, we cannot use it to determine whether the protein is intact or not. This is because the membrane proteins give relatively poor coverage in MS with seq coverage <70% being quite normal. This poor coverage is explained in part by the unusual properties of membrane proteins. The membrane region is characterized by very hydrophobic peptides and a low number of positively charge Lys/Arg. These hydrophobic peptides have a low detection limit due to aggregation and because they ionize inefficiently in electrospray ionization.

Repeating the point above: We use the sample only to to obtain cryo-EM maps and these maps are of sufficient quality to build the CTL1 model. this model that overlaps with AF predictions and the model from the publication from Xie et al., 2022. EM Maps of our sample and the Xie et al sample both show the extracellular domain and the models remain in the same conformation, with an RMSD of 0.8 Å (Figure S5B).

Reviewer #3

The revisions by the authors have improved the manuscript and it is now publishable in my opinion. I am satisfied with the clarifications regarding structural biology, and agree with the inclusion of the pH dependent uptake assays for CTL1, PNS1 and Cher1- this adds weight to the suggestion that CTL1 does not function as previously thought.

We thank the reviewer for their support and continued input toward improving the manuscript.

The title could still be improved. The FLVCR family are bona fide choline transporters, so probably shouldn't be called "proposed".

We agree. To continue to have a title with less than 100 characters, we have now adjusted it to:
“*Structural and biochemical comparison of the FLVCR and CTL membrane protein families in eukaryotes*”

with the table of content (eTOC) expansion:

“*Structure-function analysis supports that the FLVCR membrane protein family act as choline transporters, while a non-transport role is suggested for the CTL membrane protein family, previously proposed to be choline transporters.*”

Some small errors remain that should be fixed, e.g. methionine is spelled incorrectly in Figure 1.

We have corrected this typo.

Abstract- line 26 should read "appears"

We have corrected this typo.

April 30, 2026

RE: Life Science Alliance Manuscript #LSA-2025-03583RR

Dr. Bjørn Panyella Pedersen
Aarhus University
Department of Molecular Biology and Genetics
Universitetsbyen 81
MBG-AU
Aarhus C, Danmark 8000
Denmark

Dear Dr. Pedersen,

Thank you for submitting your Research Article entitled "Structural and biochemical comparison of the FLVCR and CTL membrane protein families in eukaryotes", and for your patience with our formatting requests. It is a pleasure to let you know that your manuscript is now accepted for publication in Life Science Alliance. Congratulations on this interesting work.

Your article will publish open access upon publication under a CC-BY license.

DISTRIBUTION OF MATERIALS:

Again, congratulations on a very nice paper. I hope you found the review process to be constructive and are pleased with how the manuscript was handled editorially. We look forward to future exciting submissions from your lab.

Sincerely,

Sarita Hebbar, PhD
Scientific Editor
Life Science Alliance
<http://www.lsajournal.org>